# The competition between fracture nucleation, propagation and coalescence in dry and water-saturated crystalline continental upper crust

Jessica A. McBeck[1], Wenlu Zhu[2], François Renard[1,3]

[1]Njord Centre, Department of Geosciences, University of Oslo, Norway
[2]Department of Geology, University of Maryland, College Park, U.S.A.
[3]University Grenoble Alpes, University Savoie Mont Blanc, CNRS, IRD, IFSTTAR, ISTerre, France

*Correspondence to*: Jessica McBeck (j.a.mcbeck@geo.uio.no)

**Abstract.** The continuum of behavior that emerges during fracture network development may be categorized into three endmember modes: fracture nucleation, isolated fracture propagation, and fracture coalescence. These different modes of fracture growth produce fracture networks with distinctive geometric attributes, such as clustering and connectivity, that exert important controls on permeability and the extent of fluid-rock interactions. To track how these modes of fracture development vary in dominance throughout loading toward failure, and thus how the geometric attributes of fracture networks may vary under these conditions, we perform in situ X-ray tomography triaxial compression experiments on low porosity crystalline rock (monzonite) under upper crustal stress conditions. To examine the influence of pore fluid on the varying dominance of the three modes of growth, we perform two experiments under nominally dry conditions and one under water-saturated conditions with 5 MPa pore fluid pressure. We impose a confining pressure of 20-35 MPa and then increase the differential stress in steps until the rock fails macroscopically. After each stress step of 1-5 MPa we acquire a three-dimensional (3D) X-ray adsorption coefficient field from which we extract the 3D fracture network. We develop a novel method of tracking individual fractures between subsequent tomographic scans that identifies whether fractures grow from the coalescence and linkage of several fractures or from the propagation of a single fracture. Throughout loading in all of the experiments, the volume of preexisting fractures is larger than those of nucleating fractures, indicating that the growth of preexisting fractures dominates the nucleation of new fractures. Throughout loading until shortly before failure in all of the experiments, the volume of coalescing fractures is smaller than the volume of propagating fractures, indicating that fracture propagation dominates coalescence. Immediately preceding failure, however, the volume of coalescing fractures is at least double the volume of propagating fractures in the experiments performed at nominally dry conditions. In the water-saturated sample, in contrast, although the volume of coalescing fractures increases during this stage preceding failure, the volume of propagating fractures remains dominant. The influence of stress corrosion cracking associated with hydration reactions at fracture tips and/or dilatant hardening may explain the observed difference in fracture development under dry and water-saturated conditions.

## 1. Introduction

Fracture and fault networks develop through the nucleation of new fractures, the propagation of new and preexisting fractures, and the coalescence of neighboring fractures (e.g., *Tapponnier & Brace*, 1976; *Nemat-Nasser & Horii*, 1982; *Atkinson*, 1984; *Olson*, 1993; *Lockner et al.*, 1991; *Reches & Lockner*, 1994; *Martin & Chandler*, 1994; *Kawakata et al.*, 1997; *Mansfield & Cartwright*, 2001; *Crider & Peacock*, 2004; *Jackson & Rotevatn*, 2013). Formulations of linear elastic fracture mechanics (LEFM) can describe the potential of propagation of one or a few fractures within linear elastic material (e.g., *Griffith*, 1921; *Irwin*, 1957). However, such analytical formulations struggle to describe the coalescence behavior of fracture networks as they transition from distributed, disperse networks comprised of many isolated, small fractures to more localized networks comprised of well-connected, larger fractures. This transition includes a continuum of fracture development that may be divided into three endmember modes of fracture growth: 1) nucleation, 2) isolated propagation and 3) coalescence.

The aim of this work is to provide experimental constraints on the stress and fluid conditions that promote the dominance of one mode of fracture network development over another. Identifying which of these modes dominates the others under varying conditions may be critical for accurate assessment of fracture network development. For example, if nucleation is the dominant mode of fracture development rather than isolated propagation, then using metrics that identify sites of potential fracture nucleation may be more accurate than using metrics that predict the conditions under which a preexisting fracture will grow. Metrics that indicate regions in which fractures may nucleate include the strain energy density, maximum Coulomb stress, maximum magnitude of shear stress, and highest tensile stress or least compressive stress (e.g., *Jaeger et al.*, 1979; *Atkinson*, 1987; *Du & Aydin*, 1993). Previous analyses have used some of these metrics to predict the direction of fracture growth from a preexisting fracture tip (e.g., *Olson & Cooke*, 2005; *Okubo & Schulz*, 2005; *Fattaruso et al.*, 2016). However, these metrics can lead to conflicting predictions about both the sites of new fracture nucleation and the direction of fracture growth (e.g., *Madden et al.*, 2017; *McBeck et al.*, 2017, 2020). If preexisting fracture propagation is the dominant mode of development rather than fracture nucleation, then metrics that determine the conditions under which preexisting fractures will grow, such as the critical stress intensity factor (*Isida*, 1971), and the direction of fault growth, such as Coulomb shear stress, tensile stress, and energy optimization (e.g., *Pollard & Aydin*, 1988; *Müller,* 1988*; Mary et al.*, 2013; *Madden et al.*, 2017; *McBeck et al.*, 2017), may provide more accurate predictions of fault network development than nucleation criteria. Thus, determining which mode dominates deformation under varying confinement and fluid conditions may help identify analyses suitable for successful prediction of fracture network development.

The mode of fracture growth that dominates deformation may also influence the permeability of the network and effectivity of fluid-rock interactions because these modes can control the connectivity, tortuosity, and total fracture surface area of the network (e.g., *Hickman et al.*, 1995). In particular, if a fracture network is dominated by many isolated fractures that propagate independently, it may host lower connectivity, greater tortuosity, and higher fracture surface area available for chemical reactions than a network dominated by several connected fractures that form via coalescence. The connectivity, tortuosity, and available fracture surface area may influence the effective permeability, and rate and extent of fluid-rock

interactions (*Hickman et al*., 1995; *Blanpied et al*., 1998; *Lamy-Chappuis et al*., 2014; *Frery et al*., 2015). In particular, fluid-rock interactions in the rock with many distributed small fractures that hosts greater fracture surface area may be more effective than in one with a few large fractures with lower surface area, depending on the permeability of the rock and whether the reaction is diffusion-controlled (e.g., *Renard et al*., 2000). A distributed fracture network comprised of many unconnected fractures may produce lower permeability than a more localized and connected fracture network. This difference in permeability may then influence the ability of fluid to access rock surfaces and react with them. In turn, reactions that dissolve the host rock or precipitate new material can influence the porosity and permeability (e.g*., Sausse et al.,* 2001; *Tenthorey et al*., 2003; *Lamy-Chappuis et al*., 2014). Thus, identifying the conditions under which coalescence or isolated propagation dominates may help assess the efficiency of geothermal energy and unconventional fossil fuel productions, and identify sites ideal for waste disposal or $CO_2$ sequestration (e.g., *Saeedi et al*., 2016; *Cui et al*., 2018).

To investigate the relative contributions of the three endmember deformation modes to fracture network development, we quantify the evolution of 3D fracture networks in monzonite rock samples undergoing brittle failure using in situ dynamic X-ray synchrotron microtomography. We conducted three triaxial deformation experiments at room temperature and confining pressures of 20-35 MPa. In two of the experiments, the sample was deformed at nominally dry conditions. In the third experiment, the sample was saturated with deionized water and deformed at a constant pore fluid pressure of 5 MPa under drained conditions. During the deformation tests, the maximum principal (compressive) stress $\sigma_1$ was increased in distinct steps of 1-5 MPa while the intermediate and minimum principal stresses, $\sigma_2 = \sigma_3 = 20 - 35$ MPa, were kept constant until macroscopic failure occurred (**Figure** 1). After each differential stress ($\sigma_1$ - $\sigma_2$) increase, we acquired a microtomographic scan of the deforming rock at in situ stress conditions. From these scans, we obtained the evolving three-dimensional (3D) fracture networks within the samples (**Figure** 2). We developed novel methods of tracking the growth of fractures that enable distinguishing between fractures that grow via isolated propagation and those that grow from the coalescence of several fractures. These new methods enable quantitatively comparing the competing influences of 1) nucleation and preexisting propagation, 2) isolated propagation and coalescence, and 3) local stress perturbations. Our analyses show that these competitions evolve toward macroscopic failure and depend on the stress states and interstitial fluid.

## 2. Methods

### 2.1. In situ X-ray tomography

We performed three triaxial deformation experiments with in situ dynamic X-ray synchrotron microtomography at beamline ID19 at the European Synchrotron and Radiation Facility (ESRF). We deformed monzonite cylinders 1 cm in height and 0.4 cm in diameter using the HADES apparatus (*Renard et al*., 2016). Monzonite is an igneous crystalline rock with similar mechanical properties to granite. Using the porosity measured in the tomograms, the initial porosity of each rock core is close to zero. This monzonite has a mean grain size of 450 μm (*Aben et al*., 2016). The large grain size relative to the sample size may cause the representative elementary volume (REV) of the system to approach the size of the sample. The question of the

appropriate REV size is critical to address in order to aid reproducibility of the results, but difficult to estimate. Whether or not a REV exists depends on the rheology: for elastic materials it may exist, but for softening materials it may not (*Gitman et al.*, 2007). Due to the large grain size relative to the sample size, we are above the minimum limit of a REV in granular materials (10 grains), and below the upper limit for stick-slip phenomena with glass beads ($10^7$) (*Evesque & Adjemian*, 2002). The following analysis, and previous work describing these experiments (*Renard et al.*, 2018, 2019b), find general similarities in fracture network development in these three experiments, suggesting the reproducibility and robustness of the results.

In each experiment, we imposed a constant confining pressure ($\sigma_2 = \sigma_3$) and then increased the axial stress ($\sigma_1$) in steps until the rock failed macroscopically (**Figure** 1). After each differential stress increment, we acquired a scan of the sample at in situ stress conditions with 6.5 μm voxel resolution (*Renard et al.*, 2016). The duration of each scan is within 2 minutes. The experiments were conducted at room temperature, at three different confining pressures: 20 MPa (experiment #3), 25 MPa (#5), and 35 MPa (#4). Macroscopic failure occurred in a sudden stress drop. The final scan was taken at a differential stress very close to the failure stress, typically <0.5 MPa below the failure stress. Experiments #3 and #5 were conducted at nominally dry conditions, while the sample was fully saturated in experiment #4. This sample was submerged in deionized water for 24 hours under vacuum before the experiment to help ensure that the pore space was saturated. In experiment #4, a constant pore fluid pressure of 5 MPa was maintained using two pore pressure pumps connected at each end of the sample (top and bottom). Experiment #4 is also unique in that we reached the axial stress limit of the device (200 MPa) preceding macroscopic failure, and thus we reduced the confining pressure in steps of 1 MPa from 35 to 31 MPa until the core failed. Consequently, the sample experienced 35 MPa of confining pressure for 60 scans and stress steps, and then experienced 34 MPa, 33 MPa, 32 MPa, and 31 MPa confining pressure in the final four scans preceding failure, respectively. *Renard et al.* (2018, 2019b) describe the experimental conditions in further detail. *Renard et al.* (2018) describe experiments #3 and #4. *Renard et al.* (2019b) analyze experiment #5. In the present study, we develop a new technique to follow the dynamics of fracture growth by categorizing this growth into three endmember modes of growth. The X-ray tomography data of the three experiments are publicly available (*Renard*, 2017, 2018).

## 2.2. Extraction of the fracture networks

From the time series of 3D adsorption coefficient fields acquired throughout loading, we identify fractures and pores using a standard thresholding technique. The histogram of grey-scale values from a tomogram of a porous rock tends to have two maxima indicative of the modes of the solid and air (or deionized water) populations, respectively (e.g., *Renard et al.*, 2019a). The local minimum of this histogram then determines the threshold that indicates whether voxels are identified as pore space or solid. Segmenting the tomograms with this procedure yield 3D binary fields of zeros and ones that indicate whether a voxel is within or outside of a fracture or pore. Because we employ the same threshold throughout loading in each experiment, the choice of the threshold has a similar effect for the entire time series of scans.

From the binary field, we extract individual fracture or pore objects by identifying groups of voxels that have 26-fold connectivity, the highest degree of connectivity in 3D. For each group of voxels, we calculate the covariance matrix and

corresponding eigenvectors and eigenvalues, which describe the shape of each fracture using three principal orthogonal length scales corresponding to the eigenvectors. If the pore had an ellipsoidal shape, the three eigenvalues would represent the lengths of the three axes of the ellipsoid. We then use these eigenvalues to characterize the dimensions of fractures and pores in subsequent analyses. For all of the calculations using the fracture volume, we use the actual volume of the group of connected voxels. For all calculations that depend on the placement of the fractures in space, we use the three eigenvalues calculated from the covariance matrix. Because mineral grain boundaries do not exert a significant impact on the geometry of fractures in these monzonite cores, the three eigenvectors of the covariance matrix provide a close approximation of the true fracture.

## 2.3. Identifying nucleating, propagating and coalescing fractures

After identifying the individual fractures at each loading step of an experiment, we now track the fractures across several loading steps. In addition, we develop a method that links one or more fractures at the previous loading step ($t_n$) to the next loading step ($t_{n+1}$) (**Figure** 3). This development is the central difference between this new method and the previous method of tracking fractures in X-ray tomography data developed by *Kandula et al*. (2019), and used in *McBeck et al*. (2019a). The previous method did not allow linking more than one fracture in $t_n$ to a fracture in $t_{n+1}$. Thus, *Kandula et al*. (2019) could identify when an individual fracture gained or lost volume from one loading step (and tomogram) to the next. However, this analysis could not differentiate between fractures that gained volume because one fracture propagated and opened, or because several fractures propagated and linked with each other (i.e., coalesced).

We developed this new method of tracking fractures in order to examine the competing influence of fracture coalescence and isolated propagation (**Figure** 3). Our method identifies one or more fractures in $t_n$ and one fracture in $t_{n+1}$ by searching for fractures in $t_n$ that are within five voxels of a fracture in $t_{n+1}$. We use the ellipsoidal approximations of the fractures to do this search. The limit of five voxels helps ensure that the algorithm identifies fractures that have shifted in space due to deformation. The appropriate value of this limit may differ in rocks that experience differing axial and radial strains in each loading step to those observed here. We only perform the analysis for fractures with volumes >100 voxels. This volume threshold helps exclude noise from the analysis. The appropriate volume threshold is likely different for rocks that host differing ranges of fracture volumes than those observed here. Varying the volume threshold from 100 to 500 voxels does not change the main trends described in the results (**Figure** S1).

Determining whether a fracture is nucleating, propagation or coalescing at a given time step depends on the spatial resolution of the tomogram and the amount of opening that the fracture accommodates. We may only detect fractures with apertures greater than the scan voxel size (6.5 μm). Thus, the nucleating classification refers to newly identified fractures with apertures > 6.5 μm, which may have formed in a previous loading step with apertures < 6.5 μm.

## 3. Results

### 3.1. Macroscopic mechanical behavior

The global mechanical behavior captured in the differential stress and axial strain relationships indicates that the monzonite samples undergo the deformation stages typical for brittle materials under triaxial compression (e.g., *Paterson & Wong*, 2005). We may separate the macroscopic deformation behavior into four different stages (**Figure** 1). Stage I is the initial non-linear stage corresponding to closure of preexisting defects. Stage II includes a quasi-linear relationship between stress and strain. Stage III occurs when deformation behavior deviates significantly from linearity. The yield point marks the boundary between stages II and III. Stage IV occurs shortly before macroscopic failure, when the effective elastic modulus is near zero (**Figure** 1). **Figure** 1 shows the axial strains when the initial shallowing occurs, which we refer to as the yield point in the subsequent text. We identify the yield point using the largest axial strain at which the difference between the observed differential stress and the differential stress predicted from a linear fit is less than 1% of the observed differential stress (**Figure** S2). We note that we leave the timing of the transitions from stage I to II and from stage III and IV as only qualitative in the subsequent analysis, while the transition from stage II to III is more precisely defined as the yield point. The macroscopic failure of the rocks occurred in a sudden stress drop that either completely crushed the core (experiments #3 and #5), or allowed partial recovery of the core (#4). The macroscopic failure of experiment #4 included the formation of a system-spanning fracture network oriented approximately 30° from $\sigma_1$ (Figure 4 in *Renard et al.*, 2018).

### 3.2. Fracture nucleation and preexisting fracture propagation

Here we assess the dominance of fracture nucleation relative to the growth of preexisting fractures throughout loading in the three experiments (**Figure** 4). We track the number and total volume of fractures identified in a loading step that did (i.e., preexisting) and did not (i.e., nucleating) grow from a preexisting fracture identified in the previous loading step. In this and subsequent analyses, data reported for the time closest to macroscopic failure reflect the fracture network development that occurs from the second to last ($t_{f-2}$) and final ($t_{f-1}$) scan acquired in the experiment, where $t_f$ is the time of macroscopic failure.

Throughout stages I-II in each experiment, both the number and total volume of preexisting and nucleating fractures increase with increasing strain at comparable levels (**Figure** 4). We consider the rate of growth as the increase in number or volume of fractures per strain increment. An increase/decrease in rate of growth thus marks an acceleration/deceleration in fracture growth in terms of number or volume. During the transition from stage II to III at yielding, the number and volume of the preexisting fractures accelerate, whereas the number and volume of nucleating fractures do not accelerate as quickly. Due to this bifurcation in acceleration, the number and volume of preexisting fractures exceed those of the nucleating fractures at the end of stage III and through stage IV, prior to failure (**Figure** 4a, b). At the end of stage IV, the volume of preexisting fractures exceeds the volume of newly nucleating fractures by several orders of magnitude (**Figure** 4b, c). In particular, at the end of stage VI the volume of newly nucleating fractures is 1%, <1%, and 13% of the volume of preexisting fractures in

experiments #3, #5 and #4, respectively. Overall, preexisting fracture propagation dominates fracture nucleation in the monzonite rocks deformed to failure.

Our results show that while the acceleration in the number of preexisting fractures coincides with the yield point, the acceleration in the volume of preexisting fractures becomes significant only during stage IV, when macroscopic failure is imminent. This trend may also occur for the nucleating fractures, but the number of nucleating fractures identified near the yield point is too low to draw the conclusion with confidence. Finally, the function of preexisting fracture volume relative to axial strain is approximately constant in linear-log strain-volume space (**Figure** 4c), indicating an exponential increase in total

volume as a function of axial strain. The exponents of the best-fit exponential functions of the preexisting fracture volume relative to axial strain range from 725-2000 for the three experiments, with $R^2$ values between the best-fit functions and the data of 0.85-0.98.

### 3.3. Isolated fracture propagation and fracture coalescence

To assess the influence of isolated fracture propagation relative to coalescence on fracture network development, we

develop a method to recognize when fractures develop from the merger of two or more fractures (i.e., coalesce) or from the lengthening, opening or closing of only one fracture (i.e., isolated propagation). **Figure** 5 shows the number and total volume of fractures identified as developing from two or more fractures (i.e., coalescing) or from only one preexisting fracture (i.e., propagating). We use the short-hand term *propagating* to indicate fractures that grow in isolation, but we note that fractures identified as coalescing also propagate before or while they merge.

The number of propagating fractures is larger than the number of coalescing fractures throughout loading in each experiment (**Figure** 5a). The number and volume of the propagating fractures accelerate throughout stages II-IV. In contrast, the number and volume of the coalescing fractures only appear to accelerate following yielding, throughout stages III-IV. Overall, the differences in number and volume of propagating and coalescing fractures grow larger during stages I-III.

At the end of stage IV, immediately preceding macroscopic failure, the total volume of coalescing fractures exceeds the

total volume of propagating fractures in the nominally dry experiments (experiments #3 and #5). During this stage, the volume of propagating fractures is 44% or 23% of the volume of coalescing fractures for experiments #3 and #5, respectively. In contrast, in the water-saturated experiment (#4), the total volume of coalescing fractures never exceeds the total volume of propagating fractures. Immediately preceding macroscopic failure, the volume of propagating fractures is about seven times higher than the volume of coalescing fractures in this experiment. Thus, water-saturated conditions and higher confining stress

appear to promote fracture propagation and suppress coalescence.

### 3.4. Disperse and localized fracture growth

To characterize the influence of localization and stress perturbations on fracture network development, we identify the fractures that are gaining and losing volume from one loading step to the next, i.e., growing or closing, and whether they are located near or far from another fracture. Analytical formulations of LEFM with the stress intensity factor suggest that fractures

perturb their local stress field to a distance on the order of their length (e.g., *Chinnery & Petrak*, 1968; *Segall & Pollard*, 1980; *Atkinson*, 1987; *Scholz et al*., 1993; *Davy et al*., 2010, 2013). A corollary of this concept is that fractures that are within one fracture length of other (perturbing) fractures may be more likely to grow, and less likely to close. This behavior will only be true if the local stress perturbation is favorable for growth. In contrast, local stress perturbations can also produce stress fields that hinder fracture growth, i.e., stress shadows. In this case, if a fracture lies in a stress shadow, it should be less likely to

grow, and perhaps more likely to close. We test these inferences here. In particular, we track the number of growing and closing fractures that do (i.e., near) and do not (i.e., far) have other fractures within one fracture length of them at each stress step (**Figure** 6). For example, if one fracture (fracture #1) is located within $y$ distance of another fracture (#2) with length $y$, then fracture #1 is counted in the *near* category.

The number of growing fractures matches the number of closing fractures in stages I-II early in loading (**Figure** 6a).

During stage III after yielding, the number of growing fractures accelerates while the number of closing fractures remains at similar values. The number of growing fractures that are located near other fractures (within a fracture length of them) increases with loading (**Figure** 6b). In contrast, the number of growing fractures that are far from other fractures remains roughly constant throughout loading. These varying trends produce two patterns of fracture growth before and after the yield point. In stages I-II before yielding, the number of growing fractures located far from other fractures exceeds or is similar to the number

of growing fractures located near other fractures. These observations suggest that fractures located closer to other fractures are not more likely to grow than fractures spread further apart, indicating that stress concentrations produced by developing fractures do not appreciably influence fracture development preceding yielding. In stages III-IV after yielding, however, the number of growing fractures located near to other fractures increasingly exceeds the number of growing fractures located far from other fractures. At the end of stage IV immediately preceding macroscopic failure in all three experiments, the number

of growing fractures located near to others is 3-5 times higher than the number growing far from others. When macroscopic failure becomes imminent, the stress concentrations produced by growing fractures appear to promote growth rather than suppress it.

The evolution of the number of growing fractures located near to others further highlights the influence of coalescence on fracture network development (**Figure** 6b). The number of these fractures decreases in the final loading steps just before failure

in the dry experiments (#3 and #5). In contrast, the number of these fractures continually increases in the water-saturated experiment (#4). Fracture coalescence reduces the total number of fractures as many smaller fractures merge into a few larger fractures.

## 4.  Discussion

### 4.1. The competition between fracture nucleation and preexisting fracture propagation

In these monzonite rocks undergoing brittle failure, preexisting fracture propagation dominates fracture nucleation after yielding (**Figure** 4). At a macroscopic scale, many of the conditions that favor fracture nucleation also favor preexisting

propagation, such as higher differential stress and/or lower effective confinement. At a more local scale, mechanical heterogeneities control the location of fracture nucleation and the growth of preexisting fractures. For example, *Tapponnier & Brace* (1976) documented that fracture development initiates along grain boundaries and healed transgranular fractures in granite, and new transgranular fractures propagate only at higher differential stresses. In granular cohesive rocks, such as sandstone, shear and/or tensile stress concentrations at the boundary of grains can also promote fracture nucleation (e.g., *Menéndez et al.*, 1996; *Baud et al.*, 2004; *Zhu et al.*, 2010). These mechanical controls influence the ability of fractures to nucleate and propagate following nucleation: fractures can arrest at grain boundaries and mechanical contacts, depending on the degree of stress transfer across such interfaces (e.g., *Tapponnier & Brace*, 1976; *Cooke & Underwood*, 2001; *McBeck et al.,* 2019a, b). Thus, the competing influence of these modes of fracture network development (nucleation or preexisting propagation) is difficult to predict in rocks that include such mechanical heterogeneities, and may be more challenging in rocks without such strong heterogeneities, such as monzonite. Granular rocks may contain mechanical heterogeneities that concentrate shear and/or tensile stresses more effectively than monzonite, which consists of an interlocking crystalline structure with more homogeneous mechanical properties. For example, numerical discrete element method models of sandstone indicate that the degree of strength heterogeneity between grain boundaries and intragranular material controls the proportion of fractures that nucleate at grain boundaries and those that nucleate within grains (*McBeck et al.,* 2019b). Thus, in a given sandstone volume there will likely be a greater number of sites of significant stress concentrations than in a monzonite or granite volume, and thereby a larger number of sites suitable for fracture nucleation. Consequently, we may expect a greater dominance of nucleation in sandstone and other rocks with strong strength heterogeneity than observed in these monzonite rocks.

In the crust, interfaces between mechanical sequences can exert a first order effect on the extent of fracture propagation. In sedimentary volumes consisting of parallel layers, for example, mechanical interfaces can arrest fracture growth (e.g., *Cooke & Underwood*, 2001; *Underwood et al.*, 2003). When these interfaces hinder growth in sedimentary sequences undergoing layer-parallel extension, the competition between fracture nucleation and propagation follows a systematic evolution. In these systems, new fractures nucleate, propagate perpendicular to the maximum tensile direction, open parallel to this direction, and (sometimes) arrest their propagation at an interface so that eventually the spacing between fractures is proportional to the layer thickness (e.g., *Narr & Suppe*, 1991). When the spacing between fractures reaches a certain critical value, the layer becomes saturated such that no (or few) fractures nucleate and only the preexisting fractures open in order to accommodate the applied extension (e.g., *Wu & Pollard*, 1995; *Zheng et al.*, 2019). Thus, early in loading fracture nucleation dominates, and later in loading preexisting fracture development dominates.

Here, we document how the competition between fracture nucleation and preexisting development evolves with increasing differential stress in triaxial compression (e.g., **Figure** 7), similar to the evolution observed in extending layered sedimentary sequences. Our results indicate that increasing differential stress promotes the dominance of preexisting fracture development rather than nucleation. As the fractures lengthen and open under increasing differential stress, the stress intensity factors at their tips increase (*Isida*, 1971) and thereby further promote propagation. As deformation localizes among several larger

fractures, the energetic cost of propagating preexisting fractures may become less than the cost of nucleating new fractures (e.g., *Del Castello & Cooke*, 2007; *Herbert et al*., 2015). Our data support these predictions from the linear elastic fracture mechanics and energy optimization.

## 4.2. The competition between isolated fracture propagation and coalescence

Tracking the volume of fractures that coalesce from several fractures and those that propagate in isolation without merging indicates that isolated propagation dominates coalescence throughout most of the deformation process preceding macroscopic failure (**Figure** 5). Preceding macroscopic failure, our results suggest that the presence of fluid and magnitude of confining stress may affect the competition between isolated propagation and coalescence. We deformed the water-saturated sample (experiment #4) with the highest effective confining stress; the confining pressure minus pore fluid pressure was 30 MPa. In 295    this experiment, the total volume of coalescing fractures was <10% of the volume of propagating fractures immediately preceding macroscopic failure (**Figure** 5b). In contrast, in the experiments deformed at lower confining stress (20 and 25 MPa in experiments #3 and #5, respectively) and dry-conditions, the total volume of coalescing fractures was at least twice the volume of the propagating fractures preceding failure. This difference in behavior suggests that dry conditions and lower confining stress promote coalescence rather than isolated propagation.

Many observations indicate that the magnitude of confining stress influences fracture development. In the endmember case when a rock undergoes uniaxial compression (i.e., zero confinement), experiments show that opening mode and tensile failure dominate deformation with little evidence of shear deformation (e.g., *Lin et al*., 2015). *Tapponnier & Brace* (1976) observed few shear fractures in triaxial experiments on Westerly granite under 50 MPa confining stress. With increasing confinement, fractures can appear to rotate from the orientation preferred under uniaxial compression conditions (parallel to 305    the maximum compression direction), toward the range of orientations predicted by the maximum Coulomb shear stress (e.g., *Mair et al*., 2002; *McBeck et al*., 2019a). Analyses often interpret such rotation to indicate an increasing dominance of shear deformation at the expense of tensile deformation. However, such apparent rotation may occur as many individual mode-I fractures link together so that the macroscopic trend of the fault is inclined relative to the maximum compression direction (e.g., *Peng & Johnson*, 1972; *Lockner et al*., 1991; *Renard et al*., 2019a). Consequently, the fracture geometry alone may not 310    indicate the relative proportion of shear and tensile deformation.

       Analysis of the moment tensors of acoustic emissions provides further insights to the relative proportion of shear and tensile deformation under varying confining stresses. Analysis of acoustic emissions during triaxial compression suggests that decreasing confining stress promotes tensile failure and opening at the expense of shear failure (e.g., *Stanchits et al*., 2006). This opening may enable greater access to preexisting fractures than shear deformation, thereby promoting the likelihood of 315    coalescence. For example, mixed-mode fractures may tend to have larger apertures than fractures dominated by shear deformation. Consequently, mixed-mode failure may result in thicker fractures that provide greater surface area to which other fractures can link than thinner fractures produced predominantly by shear. The presence of damage zones surrounding crustal faults and the decreasing of the thickness of such damage zones with depth (e.g., *Harding*, 1985) support the idea that confining

stress localizes deformation in low porosity crystalline rock. Confining stress tends to reduce the proportion of tensile deformation relative to shear deformation, and thus may localize deformation into thinner zones, in the absence of cataclastic flow and ductile deformation.

The applied confining pressure in experiment #5 was 5 MPa higher than that of experiment #3, but these two dry samples show similar proportions of fracture propagation and coalescence. Consequently, it is unlikely that the 5 MPa higher effective stress of experiment #4 compared to experiment #5 is the primary trigger of the different behaviors observed in these experiments. We suggest that the presence of water is responsible for the transition from isolated propagation to coalescence-dominated fracture network development. We acknowledge that this conclusion rests on only three experiments and further work is required for more robust support of this idea. However, previous work focused on the influence of water on fracture network growth supports this idea. In particular, this work shows that chemical reactions at fracture tips can influence fracture propagation. Such stress corrosion cracking occurs when chemical reactions reduce the fracture toughness and thereby promote crack propagation (e.g., *Anderson & Grew*, 1977). When water is present, hydrogen bond formation weakens the Si-O bond in quartz-rich sandstones, producing water-weakening (e.g., *Baud et al.*, 2000). Stress corrosion cracking may thus promote nucleation at the expense of coalescence in the water-saturated monzonite experiment.

Changes in pore fluid pressure can also affect the fracture propagation rate (*Ougier-Simonin & Zhu*, 2013, 2015). Recent studies show that at the same effective pressure and loading, fault propagation in intact serpentinite is slower in samples with higher pore fluid pressures (*French & Zhu*, 2017). When a fluid-saturated rock dilates, the pore pressure may drop and thereby reduce the local effective confinement and strengthen the rock, i.e., dilatant hardening (e.g., *Brace & Bombolakis*,1963; *Rice*, 1975; *Rudnicki & Chen*, 1988; *Ikari et al.*, 2009; *Xing et al.*, 2019; *Brantut*, 2020). This strengthening can then slow the rate of fracture propagation from dynamic to quasi-stable (*Martin*, 1980; *French & Zhu*, 2017). Dilatant hardening can operate in intact rock as well as saturated gouge zones (*Ikari et al.*, 2009; *Xing et al.*, 2019). For example, increasing pore pressure causes the frictional behavior of antigorite gouge to evolve from velocity-weakening to velocity-strengthening (*Xing et al.*, 2019). Dilatant hardening may influence fracture development in the water-saturated experiment (#4) if the evolving permeability of the network is high enough to allow fluid flow at the time scale of the experiment. Using the porosity identified in the tomograms acquired immediately preceding failure, the porosity of the rocks in each experiment ranges from 0.06% (#4), 0.2% (#3), and 1.6% (#5) at this stage. Following the relationships between porosity and permeability calculated for dynamically fractured monzonite cores (*Aben et al.*, 2020), rocks with 0.06-1.6% porosity may have permeability $10^{-16}$ to $10^{-18}$ m$^{-2}$. With this range of permeability and dimensions of the rock core, water requires about less than a minute to 45 minutes to traverse the core from top to bottom (**Text** S1). Thus, the time interval of the loading steps (3 minutes) may allow water to flow between fractures, enabling the effects of stress corrosion cracking and dilatant hardening to operate at least in the final stages preceding failure. Earlier in the experiment, when the porosity and permeability is lower, the lower flow rate may suppress such effects. Further experimental investigations are needed to distinguish between the relative importance of stress corrosion cracking and dilatant hardening on fracture development within water-saturated rocks.

These observations and our analyses suggest that the presence of water (producing stress corrosion) and high pore fluid pressure (producing dilatant hardening) promote slower, more isolated fracture network growth, rather than faster, coalescence-dominated growth. Understanding the mechanical and chemical conditions that favor one mode of fracture growth over another (e.g., fracture coalescence versus isolated fracture propagation) has important implications in many energy and environmental engineering practices. For example, when their connected porosities are comparable, fracture networks produced by the propagation and coalescence of many small fractures may have lower connectivity, higher tortuosity and lower permeability than networks consisting of a few large fractures. However, the fracture networks consisting of numerous small fractures may be more efficient in shale gas exploration and $CO_2$ sequestration (e.g., *Xing et al.*, 2018).

## 4.3. The influence of local stress perturbations on fracture growth

A clear factor in fracture network development is the fracture network density, clustering, or localization. For example, earthquakes are more likely to arrest at the ends of faults that are >5 km from another fault (*Wesnousky*, 2006). Indeed, the distance between fractures is one of the key parameters that predicts whether they grow or close from one stress step to the next in X-ray tomography triaxial compression experiments on marble, monzonite and granite rocks (*McBeck et al.*, 2019a). Analytical solutions from LEFM provide a mechanical interpretation of these observations. These solutions indicate that a fracture will perturb the local stress field to a distance on the order of their length (e.g., *Scholz et al.*, 1993). Following this idea, we may use the number of growing fractures located within this threshold to another fracture to determine if stress perturbations produced by growing fractures tend to promote or hinder growth.

Our observations suggest that local stress perturbations produced by growing fractures promote the growth of neighboring fractures during stages III-IV preceding macroscopic failure (Figure 6). During these stages, the number of fractures that grow and are located within one fracture length exceeds the number of fractures that grow and are located outside of this threshold. Preceding yielding, however, similar numbers of growing fractures are located both within and outside this threshold. When the fracture network is more diffuse under lower differential stress, the distance between fractures does not appear to influence whether a fracture grows or closes (i.e., **Figure** 7). When the fracture network becomes more clustered, the distance between fractures appears to influence whether a fracture grows or closes. Our results highlight the conditions under which stress perturbations influence growth in rocks under triaxial compression that host fracture networks with a variety of spatial distributions.

## 5. Conclusions

In situ dynamic X-ray tomography during the triaxial compression of crystalline rocks reveals the competing influence of three modes of fracture network development: 1) nucleation, 2) isolated propagation and 3) coalescence. We find that the influence of these modes evolves throughout loading, with clear transitions near yielding and macroscopic failure. Preexisting fracture propagation, including isolated propagation and coalescence, becomes the dominant mode of deformation following yielding. Coalescence then becomes the dominant mechanism of fracture network development in dry samples under lower

confinements only immediately preceding macroscopic failure. Isolated propagation remains the dominant mechanism
throughout loading in a water-saturated sample under higher confinement. Compared to the prediction that fractures promote growth by perturbing their local stress field to a distance on the order of their length (e.g., *Scholz et al.*, 1993), our observations only match these expectations in the stages of the experiments between yielding and macroscopic failure. Preceding yielding, however, the fractures that are growing are not significantly closer to other fractures, indicating that their stress perturbations do not promote the growth of neighboring fractures. When the rock experiences lower differential stress and the fracture
network is more distributed, 1) similar numbers of new fractures nucleate and preexisting fractures grow, 2) isolated propagation dominates coalescence, and 3) local stress perturbations do not appear to promote fracture growth (Figure 7). When the rock experiences higher differential stress following yielding, 1) preexisting fracture propagation dominates new fracture nucleation, 2) coalescing fracture volume exceeds the propagating fracture volume in dry samples when macroscopic failure is imminent, and 3) local stress perturbations promote fracture growth.

*Data availability*. The data are available on the Norstore repository (*Renard*, 2017, 2018).

*Author contributions*. JM and FR performed the experiments, analyzed results, and wrote the manuscript. WZ analyzed results and wrote the manuscript.

*Competing interests*. There are no competing interests.

*Acknowledgements*. We thank Elodie Boller, Paul Tafforeau, and Alexander Rack for providing advice on the design of the tomography setup, Benoît Cordonnier for experimental expertise, and Sanchez Technology for building the deformation apparatus. The Research Council of Norway (awards 272217 to FR and 300435 to JM) and U.S. National Science Foundation (EAR-1761912 to WZ) funded this work. The European Synchrotron Radiation Facility allocated beamtime (Long Term Proposal ES-295). We thank guest editor André Niemeijer, reviewer Frans Aben and an anonymous reviewer for suggestions
that improved this manuscript.

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

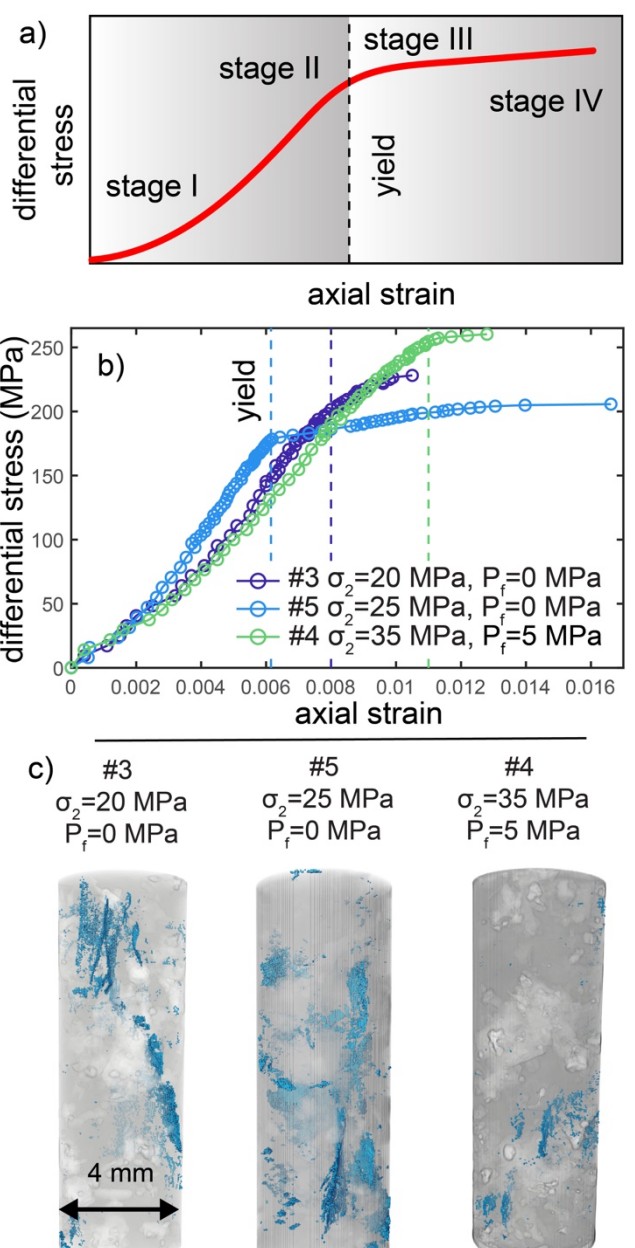

Figure 1: Macroscopic behavior of each experiment produced by fracture network development. a) Macroscopic stages of deformation. Stage I is the initial non-linear stage corresponding to the closure of preexisting defects. Stage II includes the quasi-linear relationship between stress and strain. Stage III occurs when deformation behavior deviates significantly from linearity. The yield point marks the boundary between stages II and III. Stage IV occurs close to macroscopic failure, when the effective elastic modulus is near zero. The timing of the transition between stages I-II, and stages III-IV remains approximate in this analysis. b) Differential stress and axial strain relationships of the three experiments: #3, #4, and #5. Circles show the conditions when a tomogram was acquired. The applied confining stress and pore fluid pressure increase from monzonite #3 ( $\sigma_2 = 20\ MPa, P_f = 0$ ), #5 ( $\sigma_2 = 25\ MPa, P_f = 0$ ) and #4 ( $\sigma_2 = 35\ MPa, P_f = 5\ MPa$ ). c) Fracture geometry in the final scan in all three experiments. Fractures shown in blue, minerals shown with transparent grey and white. The fracture network geometry in the last scan acquired before macroscopic failure includes longer, more volumetric, and more connected fractures in the experiments with $\sigma_2 = 20 - 25\ MPa$ and $P_f = 0$ (#3, #5) than in the experiment with $\sigma_2 = 35\ MPa, P_f = 5\ MPa$ (#4).

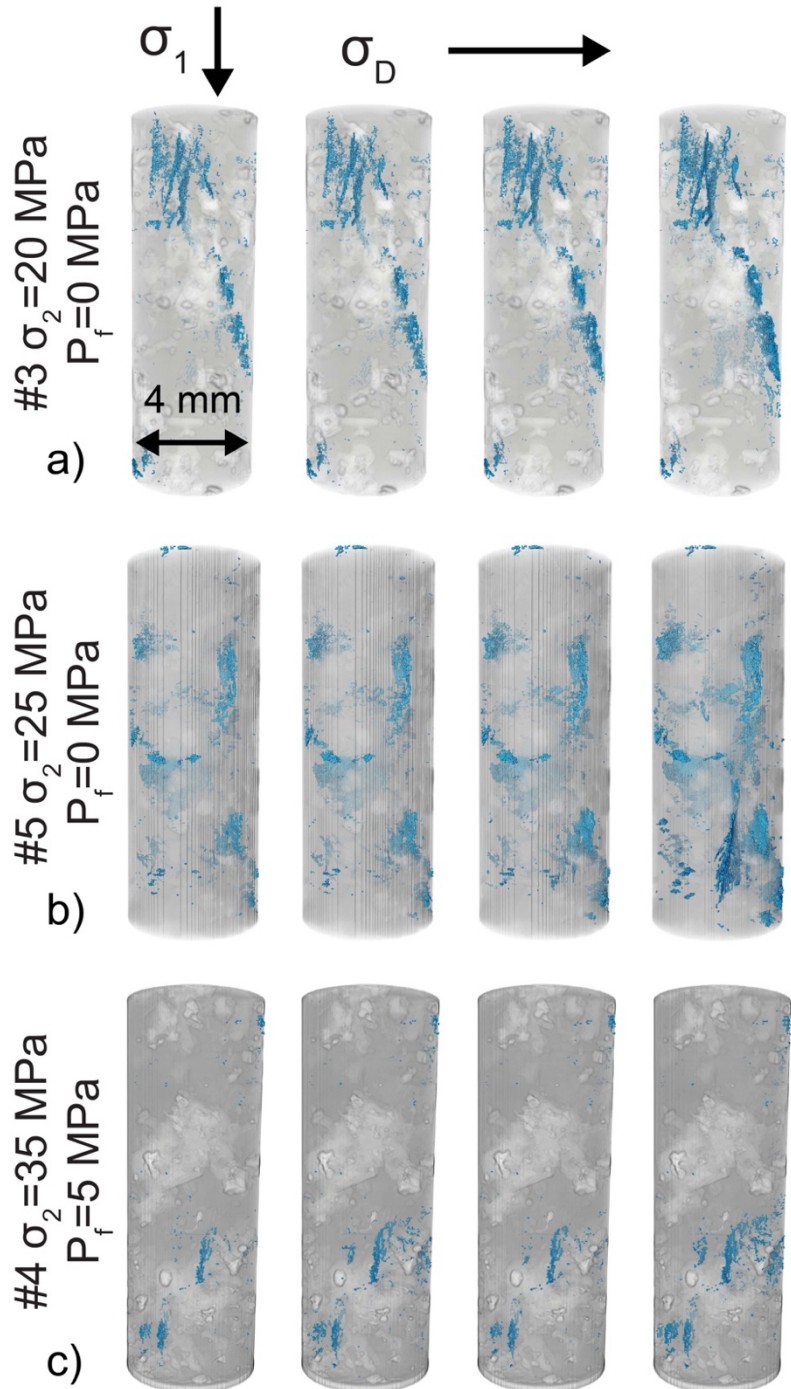

**Figure 2: Evolving fracture networks in the final four loading steps of each experiment before system-size failure.**

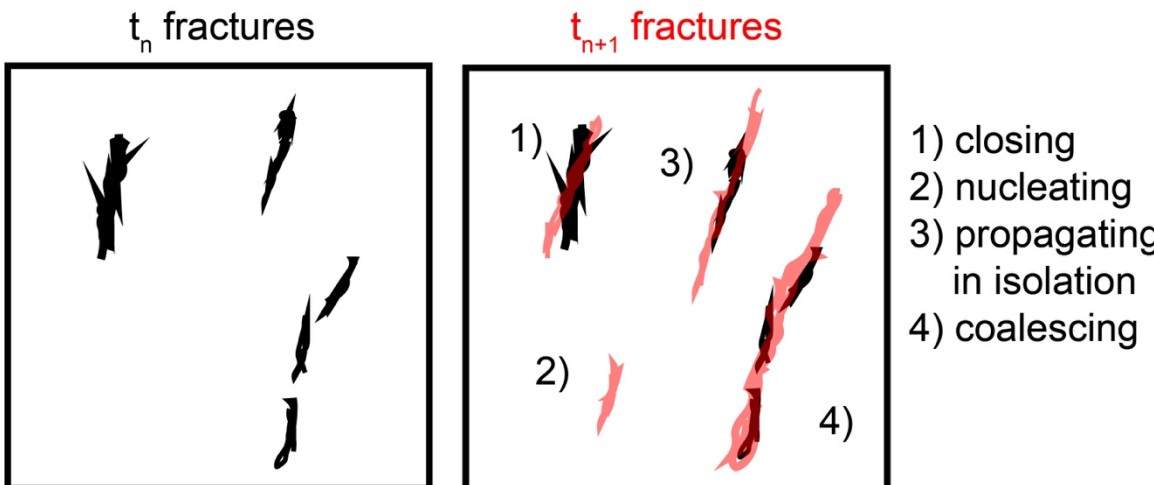

**Figure 3: Modes of fracture network development captured by algorithm. By tracking individual fractures in sequential scans, we can identify fractures that 1) close, 2) nucleate, 3) propagate in isolation and 4) coalesce from one time to the next, $t_n$ to $t_{n+1}$.**

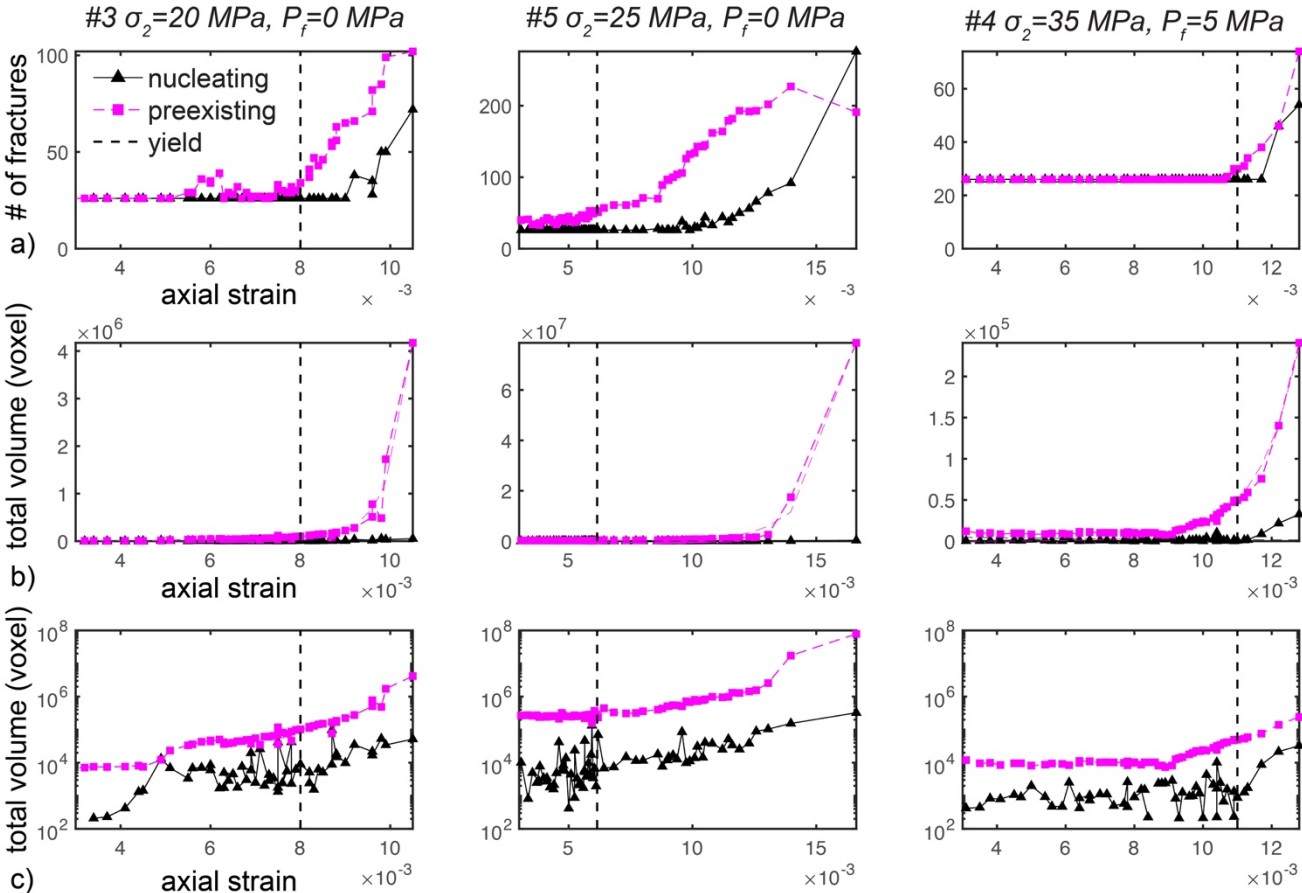

Figure 4: The competing influence of fracture nucleation and preexisting growth in each experiment. The applied effective pressure ($\sigma_2 - P_f$) increases from left to right. a) The number of fractures identified as nucleating or preexisting in each loading step. The total volume of the nucleating and preexisting fractures in linear b) and log-linear c) space. Dashed vertical lines show the axial strain at the macroscopic yielding point identified from the shallowing of the stress-strain curves (Figures 1, S2), separating stages I-II and III-IV. The pink lines without markers (b) show the best-fit exponential functions of the data. The total volume of preexisting fractures exceeds the volume of newly nucleating fractures in the final loading steps preceding macroscopic failure, indicating the dominance of preexisting development rather than nucleation. The increase of the volume of nucleating fractures after yield is more significant in the water-saturated sample compared to the nominally dry samples.

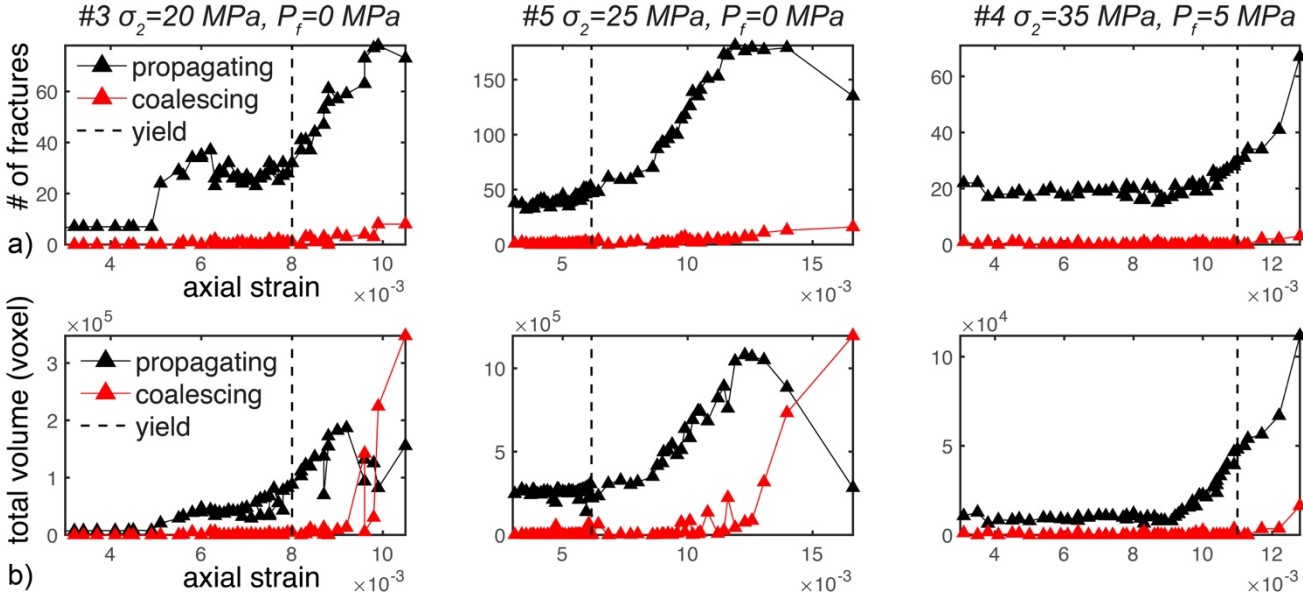

**Figure 5: The varying influence of preexisting fracture coalescence and propagation. a) The number of fractures propagating in isolation (black) and coalescing (red). b) The total volume of fractures propagating in isolation or coalescing. Prior to macroscopic failure, the total volume of propagating fractures decreases and the total volume of coalescing fractures increases in the nominally dry experiments (#3 and #5), indicating the dominance of coalescence rather than isolated propagation. In contrast, in the water-saturated experiments, the propagating fractures dominate throughout loading.**

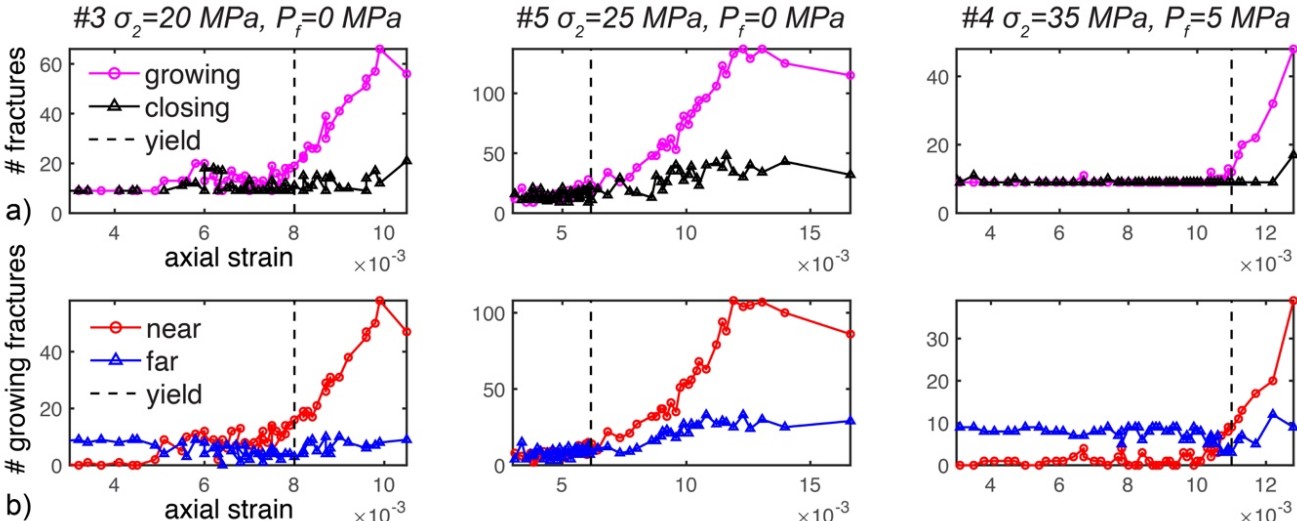

**Figure 6: The influence of stress perturbations on fracture growth. a) The number of growing (magenta circles) and closing (black triangles) fractures. b) The number of growing fractures that do (*near*, red circles) and do not (*far*, blue triangles) have other fractures within one fracture length of them throughout loading. If one fracture (fracture #1) is located within $y$ distance of another fracture (#2) with length $y$, then fracture #1 is counted in the *near* category. Following the macroscopic yield point, the number of growing fractures located near to other fractures exceeds the number located far from others. This observation suggests that the local perturbations of the stress field produced by fracture growth tend to promote the growth of other fractures following yielding.**

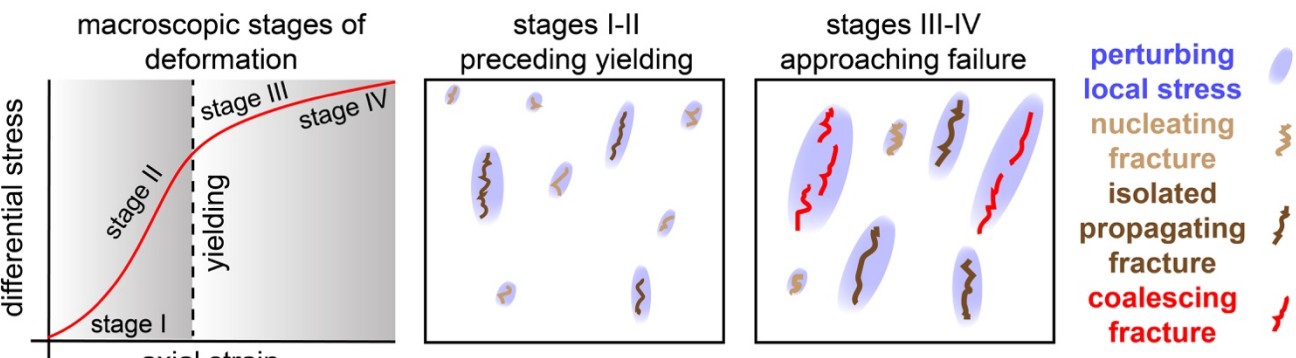

**Figure 7: Schematic of varying modes of fracture development observed preceding yielding and approaching macroscopic failure. Nucleating, propagating, and coalescing fractures shown in light brown, dark brown and red, respectively. Blue ellipsoids show the approximate extent of the perturbation of the local stress field produced by fracture network development. When the rock experiences lower differential stress and the fracture network is more distributed, 1) similar numbers of new fractures nucleate and preexisting fractures grow, 2) isolated propagation dominates coalescence, and 3) local stress perturbations do not appear to promote**

**fracture growth. When the system approaches macroscopic failure, 1) preexisting fracture propagation dominates new fracture nucleation, 2) coalescence dominates isolated propagation in the experiments with the lowest confining stress and dry conditions, and 3) local stress perturbations promote fracture growth.**