# Peer review of "The competition between fracture nucleation, propagation and coalescence in dry and water-saturated crystalline continental upper crust"

_Solid Earth, 2020_

## Referee Comment (RC1) · Franciscus Aben (Referee) · 11 Aug 2020

The manuscript entitled 'The competition between fracture nucleation, propagation, and coalescence in the crystalline continental upper crust' by McBeck et al. aims to illuminate the nucleation, growth, and coalescence of micro-fractures in crystalline rock prior to sample-size shear failure, and how this 'road-to-failure' varies in the presence/absence of pore fluids. To do so, three shear failure experiments (2 dry, 1 saturated) were conducted in a triaxial vessel, whilst obtaining a full 3D X-ray tomography model at set intervals of differential stress up to sample failure. The 3D X-ray tomography data was analysed to obtain measures for microfracture nucleation, propagation,

and coalescence. The main conclusion of the manuscript is that under fluid saturated conditions, microfractures tend to propagate more in isolation rather than coalesce with nearby microfractures relative to the dry case. This interesting conclusion based on the observations and excellent data analysis warrants publication, but the manuscript first needs to address a number of significant problems. These problems comprise the lack of clarity on the main aim of the manuscript, and a somewhat tedious discussion section with unclear/inconsistent arguments. I hope that my comments below will be of help to improve the quality and the originality of the manuscript.

Major comments:

1. The main aim of the manuscript is not clear: Is it the 'road-to-failure' with an additional step in the X-ray tomography data analysis (i.e., quantification of fracture coalescence), or is it trying to elucidate the difference in pre-failure deformation for dry and saturated conditions? This duality is making it difficult to follow, especially in the introduction and the discussion sections of the paper, and the authors may wish to rethink this. Moreover, the 'road-to-failure' has been studied and presented by (some of the) authors in other recent manuscripts, partly on the same dataset, and the additional data analysis step feels like a somewhat meager addition to these previous works. I feel that the observations on microfracture development with/without fluids does contribute more significantly to progressing our understanding of brittle rock deformation between the yield point and sample-scale failure, and so I recommend to emphasize this as the main aim of the manuscript (studied with the approach of measuring of fracture coalescence, propagation, etc.).

2. Following on this, the title does not cover entirely the content of the manuscript, and should contain some mention of the dry vs. saturated conditions.

3. I believe that (part of) the data has been presented in other manuscripts, so it should be clarified better what the innovative aspect is of this manuscript in relation to previous ones. Also, indicate which sample data sets have been presented in which manuscripts

before.

4. The methodology section 2.1 is very short and lacks some basic experimental information: What deformation rig was used, what is the voxel size, what was the axial loading rate, how was axial shortening determined, and how much time was allowed for the pore fluid pressure to equilibrate across the sample prior to the onset of loading, and in between load steps?

5. Line 145: The authors may want to add some clarification on the meaning of nucleating fractures in their data: It seems to me that the fractures that appear within the resolution of the X-ray data at each step may have been there the previous step as well, only not detected due to their size/small volume. It is most likely that they nucleated from a preexisting defect (grain boundary, cleavage plane) that initially had no volume to begin with. This does not hamper the analyses here, but it would clarify that the term nucleation used here is somewhat relative to scale/resolution, and does not describe fractures forming out of the blue. This brings up the interesting point as well on what is actually measured: The volume of microfractures. Do the authors think there are many 'hidden' pure shear microfractures without much opening (i.e., volume) in their data?

6. The authors attempt to explain coalescence from a linear elastic fracture mechanics perspective, with the hypothesis that fractures near each other are more likely to grow because their fracture tip stress concentrations interact. This is introduced first in section 3.4, and further expanded upon in section 4.4. This hypothesis is not well explained or quantified: It seems to me that such an interaction depends on the length of the fractures involved (longer fractures, larger stress intensity) and on their orientation, as well as on the exact stress fields around them (e.g., mode-II fractures have reduced and increased stresses near their tip, whereas mode-I fractures do not). So I am not sure if I understand well or agree with line 208: 'The observations match the expectations of LEFM'.

Secondly, contrary to this hypothesis is the statement in the introduction that LEFM
cannot explain well fracture coalescence (line 30-35), so why choose this as a frame-work to explain the observations on coalescence?

7. Section 4.2 discusses the competition between fracture nucleation and isolated propagation. This is somewhat tedious because the authors elect to use the analogy of sandstone deformation and models designed for layered sedimentary sequences for crystalline low porosity rock. I do not feel this is very informative: Triaxial deformation of granular aggregates is very different from low porosity rock, and the step from a small-sized crystalline rock sample to a sedimentary basin feels like a leap. Most discussion is summarized in the last paragraph of this section: This competition seems adequately explained by the fracture length dependent stress intensity factors, so that a few growing fractures shield shorter (nucleating) fractures. As a suggestion, the authors could analyse the lengths of the fractures in loading-parallel direction in their data to provide a somewhat more quantitative argument here.

8. I feel that Section 4.3 contains some similar problems as discussed above: The sandstone analogy is not a very helpful argument to explain competition between iso-lated fracture propagation and fracture coalescence in crystalline rock, and neither is fault damage zone evolution: The presented data is on an initially intact sample without a pre-existing fault zone. Further irrelevant excursions include the 316-318 on dilation in gouge materials.

9. Section 4.3 contains, in my opinion, the most interesting discussion: The influence of fluids on microfracture evolution prior to sample-sized failure. The authors first discuss the different confining pressures on all three samples as the source for the different microfracture evolution, and rule this out as a conclusion. Second, the chemical effect of water on crack propagation is discussed, followed by a discussion on the mechanical effect of a pressurized fluid. The authors conclude, rightly so, that these last two effects cannot be distinguished from each other and future research is necessary. I largely agree with the line of thought and the conclusion of the authors, but there are a few caveats and/or additional points that need to be addressed, starting at the level of the

experiment: How does the sample size (4x10 mm) influence the reproducibility of the experiments, especially given the relative large (450um on average) grain size of the material (for instance, some grains seem to have dimensions of > 1mm in the 3D CT models)?

Secondly, dilatancy hardening is presented as a mechanism to influence microfracture evolution, but how specifically is not clear. Dilatancy is often discussed on the scale of cm-size (and larger) shear fractures or fault planes, where fault roughness and microfractures around the shear plane accommodate the dilation. Here, the microfracture regime does not yet have such a centralized structure, but could it be possible that larger and coalescing microfractures have a larger dilatancy rate than smaller fractures, so that the former are more affected by dilatancy?

On stress corrosion: The authors could try to include a back of the envelop calculation on how long it takes for fluids to reach the crack tips during and after a deformation step – i.e., are the crack tips wetted during propagation, and has the pore fluid pressure equilibrated within the sample? I have measured hydraulic properties on these monzonites that may prove helpful to measure diffusion times. ("Variation of Hydraulic Properties Due to Dynamic Fracture Damage: Implications for Fault Zones", JGR 2020)

10. Maybe I have missed it, but I could not find an in-text reference to Figure 7.

11. I feel that there is some overlap in the discussion at the end of section 3.4, and the discussion on LEFM and coalescence in section 4.4. Consider cutting the part in section 3.4.

Minor comments

12. Line 31-32: I do not think this statement is correct: LEFM is scale-independent.

13. Line 38: Successful in what?

14. Line 40: Clarify what is meant with the mode of failure.

15. Line 60-61: Does this not depend on whether the reaction is diffusion or precipitation controlled?

16. Line 108: The fractures have been simplified as ellipsoids, how realistic is this shape, especially for coalesced fractures?

17. Line 122: In the description of how fractures are tracked from one X-ray dataset to the next, would it be clearer to speak of fracture volume instead of fracture?

18. Line 126: Insert 'to' in between step and those.

19. The figure references are somewhat chaotic: I would refrain from referring to the figures until the results section, and not refer to results figures in the introduction.

20. Line 138, line 227: The term elastic is not correct here, because unloading at this point would not reproduce the near-horizontal stress-strain curve.

21. Section 3.1: How did the samples look like post-failure? Did they exhibit a single shear fracture at a 30-degree angle to the loading axis?

22. Line 147: Do I understand correctly that nucleating fractures from a previous load step are counted as propagating fractures in the next step (i.e., the nucleation counter is set to zero)?

23. Line 150: Repetition from section 2, can be removed.

24. Line 159: Was the volume of fluid expulsed from the pore pressure pumps measured? If so, do they match with the volume increase inferred from microCT?

25. Line 165: Are the exponents of the increase comparable between all three samples?

26. Line 170: develop a → used our developed (the method was already explained in section 2).

27. Paragraph 184-196: The technical part of how to define near and distant fractures

should move to the method section. Also, how were closing and growing fractures defined?

28. Line 217: in during → in.

29. Line 322: The word 'analyses' in this context suggests some calculations/quantification. Maybe 'Discussion'?

30. Figure 4: Would it be possible to indicate the four deformation stages here? Showing the fracture volume in mm3 instead of voxels would make it easier for readers to extract dilation rates. In panel (c), would it be possible to have the same scale, so that the logarithmic trends are easily comparable (also for figure 5b)?

31. Some referencing is incomplete or skips over classic papers:

- line 28: The development of microfracturing with stress was already well documented by earlier studies than those cited here, especially from the 70s onward. For instance, Tapponier & Brace, 1976 have performed excellent microstructural work on this (see Paterson and Wong, section 5.7.4, for more refs).

- Line 223-224: This statement is not correct: mechanical data, AEs, and microstructures show the development of microfracture networks past the yield point; Brace, Paulding, and Scholz 1966 inferred microcracking to be responsible for significant pore volume change during loading; Tapponier & Brace, 1976 and Wong 1982 show microstructures; AEs by Scholz 1968. These are examples of the older, more classic works that show this.

- Line 316: Martin III, 1980 ("Pore pressure stabilization of failure in westerly granite") may be more relevant here, as it shows the phenomenon in crystalline rock.

---

## Referee Comment (RC2) · Anonymous Referee #2 · 12 Aug 2020

The manuscript "The competition between fracture nucleation, propagation and coalescence in the crystalline continental crust" by Jessica A McBeck, Wenlu Zhu, and Francois Renard addresses the controls of development of fracture networks. McBeck et al. present an experimental study in which the fracture network development was assessed via microtomography during triaxial mechanical tests on two dry and one water-saturated sample. The data they acquired is remarkably and the method provides a great example of how fracture networks can be tracked during loading. The main outcome, that stress state and saturated vs dry conditions of the sample are the main controls of which (endmember) fracture network develops on the way to macroscopic failure, is not reflected in the title, introduced, clearly highlighted in the methods, represented in the results or adequately discussed. The authors need to address all of the following issues so the community can appreciate the scientific contribution.

Major comments: 1. The main message of the manuscript is not clear. It is not clear if they want to highlight the methodological approach or the results they obtained by applying the method. The research question for this paper though is well hidden. The scope of the manuscript, the objectives and hypothesis are not clear. 2. It is not clear what the motivation for these experiments was. Data seems to be the same as in previous publications, which is fair to use as getting proposals funded and time allocated to do the experiments can be difficult, but it needs to made clear, where this data is new and where (re)used. 3. In the introduction, the overall concept of how fracture networks develop is not clearly outlined, thus that all assumptions and reasoning is vague. References are missing in many parts, which would allow substantiating some of the party awkward assumptions. The controlling variables which are used in the experiments and seem to be the main outcome are not introduced at all (effect of stress on fracturing, interstitial fluids). 4. The methods do not introduce the techniques applied – both the mechanical loading (e.g. rate of loading) and the tomography (e.g. which voxel size), as well as how you analyse the data (e.g. volume calculation, attribution to which mode). 5. The material used is not introduced at all. No description of the microstructure, no material properties (e.g. porosity). This makes it impossible to relate the tomography images and fracture network development to anything. The nucleation and propagation, especially at lower stress steps will be at grain boundaries and pre-existing defects and flaws. 6. The first part of the results seems to belong to the methods, yet it is not clear which point is made. The description and representation of the results are hard to follow and do not seem to grasp/show important information. For example, you could colour the "new" fractures and the ones that coalescence in the shown steps differently. 7. The structure of each section is flawed. The wording is unclear. Logical jumps make it very hard to follow the text. Especially in the introduction, the methods and results. 8. Most parts of the discussion seem to be about something completely different than the experiment (upscaling- from 10mm to upper crustal) or

Interactive
comment
research question (I am assuming crystalline rocks – yet the discussion is on sedimentary basins). The development of the fracture network is not discussed. References, if given, do not fit the topic. 9. The conclusion is contradicting the introduction in several aspects (e.g. LEFM) and is tedious as it simply repeats some statements made before which are not substantiated in the manuscript. Detailed comments: I have commented on the manuscript in detail for the Abstract and the Introduction (see supplement .pdf). The extent of these comments highlight some of the main issues of the manuscript and are alike for the following sections. The Figures are not fitting the manuscript or provide a visualization to enhance the text, some detailed comments can be found there. In addition but not exclusive for the supplement information comments are: Figure S1: - "vox" –> "voxel" - The variation in fond size and labelling position is a bit irritating. Could you work on it? - what does this # refer to? why #3, #5 and then #4. Maybe add to caption what the three panel show. - log scale hardly visible -consider using a different symbol/colour for this type to clearly distinct from the nucleation, above. - Caption: This figure does not show this. It only shows it in comparison to another figure. Please name which figure this relates to. -Caption: The main trends are not indicated (in figure or text) - what are they? To make the point, you could add the trends of the 100 voxels to the figures.

Figure S2: -Again, fond size and labelling position are a bit irritating. Why did you change the colour scheme? -Why is this yield (point) line in red, while in a) they are in the same colour as the other lines

Please also note the supplement to this comment:
https://se.copernicus.org/preprints/se-2020-114/se-2020-114-RC2-supplement.pdf

**Supplement:**

[revised manuscript text omitted]

---

## Author Comment (AC1) · 9 Sep 2020

**Dear Editor Niemeijer, Dr. Aben and anonymous reviewer,**

**Thank you for these constructive reviews. We have significantly modified the abstract, introduction and discussion to more clearly specify the motivation and new contribution of this work. We respond to the comments point-by-point below in bold font. We added numbers to each point for clarity. We respond to the annotations of the manuscript by the anonymous reviewer in the attached document.**

**Thanks,**
**Jessica McBeck**

Franciscus Aben (Referee)

The manuscript entitled 'The competition between fracture nucleation, propagation, and coalescence in the crystalline continental upper crust' by McBeck et al. aims to illuminate the nucleation, growth, and coalescence of micro-fractures in crystalline rock prior to sample-size shear failure, and how this 'road-to-failure' varies in the presence/absence of pore fluids. To do so, three shear failure experiments (2 dry, 1 saturated) were conducted in a triaxial vessel, whilst obtaining a full 3D X-ray tomography model at set intervals of differential stress up to sample failure. The 3D X-ray tomography data was analysed to obtain measures for microfracture nucleation, propagation, and coalescence. The main conclusion of the manuscript is that under fluid saturated conditions, microfractures tend to propagate more in isolation rather than coalesce with nearby microfractures relative to the dry case. This interesting conclusion based on the observations and excellent data analysis warrants publication, but the manuscript first needs to address a number of significant problems. These problems comprise the lack of clarity on the main aim of the manuscript, and a somewhat tedious discussion section with unclear/inconsistent arguments. I hope that my comments below will be of help to improve the quality and the originality of the manuscript.

Major comments:
1. The main aim of the manuscript is not clear: Is it the 'road-to-failure' with an additional step in the X-ray tomography data analysis (i.e., quantification of fracture coalescence), or is it trying to elucidate the difference in pre-failure deformation for dry and saturated conditions? This duality is making it difficult to follow, especially in the introduction and the discussion sections of the paper, and the authors may wish to rethink this. Moreover, the 'road-to-failure' has been studied and presented by (some of the) authors in other recent manuscripts, partly on the same dataset, and the additional data analysis step feels like a somewhat meager addition to these previous works. I feel that the observations on microfracture development with/without fluids does contribute more significantly to progressing our understanding of brittle rock deformation between the yield point and sample-scale failure, and so I recommend to emphasize this as the main aim of the manuscript (studied with the approach of measuring of fracture coalescence, propagation, etc.).

   **We have significantly modified the introduction and underscored throughout the text that the central focus of the paper is quantifying the competition between nucleation, propagation and coalescence, and tracking how this competition changes throughout loading (toward failure) and in dry and saturated conditions (lines 9-15, 42-43, 57-58, 76-77, 194-195, 220-221, 223). None of our previous work (or other analyses that we are aware of) have quantified these different modes of fracture growth, which exert a significant impact on permeability and fluid-rock interactions. Therefore, the main outcome of the present study is the quantification of the dominance of nucleating, propagating, and coalescing fractures. The secondary outcome is**

**the difference of behaviour in the water-saturated sample compared to the two dry samples.**

2. Following on this, the title does not cover entirely the content of the manuscript, and should contain some mention of the dry vs. saturated conditions.

   **We have now modified the title as suggested.**

3. I believe that (part of) the data has been presented in other manuscripts, so it should be clarified better what the innovative aspect is of this manuscript in relation to previous ones. Also, indicate which sample data sets have been presented in which manuscripts before.

   **The three experiments have been described in Renard et al. (2018) (#3 and #4) and in Renard et al. (2019) (#5). However, the analyses presented here provide a fundamental advance from this previous work. The new data processing done here enables quantifying the growth mode of individual fractures following a stress step increase: 1) nucleation, 2) propagation of an existing fracture, 3) coalescence, and thus tracking the evolution of these three kinds of growth modes until system-size failure. We have now clarified how this work differs from previously published work (lines 85-90, 119-121). Please see response to comment #1 above also.**

4. The methodology section 2.1 is very short and lacks some basic experimental information: What deformation rig was used, what is the voxel size, what was the axial loading rate, how was axial shortening determined, and how much time was allowed for the pore fluid pressure to equilibrate across the sample prior to the onset of loading, and in between load steps?

   **We have now added this relevant information in Section 2.1.**

5. Line 145: The authors may want to add some clarification on the meaning of nucleating fractures in their data: It seems to me that the fractures that appear within the resolution of the X-ray data at each step may have been there the previous step as well, only not detected due to their size/small volume. It is most likely that they nucleated from a preexisting defect (grain boundary, cleavage plane) that initially had no volume to begin with. This does not hamper the analyses here, but it would clarify that the term nucleation used here is somewhat relative to scale/resolution, and does not describe fractures forming out of the blue. This brings up the interesting point as well on what is actually measured: The volume of microfractures. Do the authors think there are many 'hidden' pure shear microfractures without much opening (i.e., volume) in their data?

   **We have now added the important point that the identification of nucleating fractures depends on the scan voxel size, and the identification of all of the fractures depend on their opening (lines 159-162). It is difficult to make a quantitative statement about potential hidden fractures, and thus the proportion of shear vs. dilation. Early studies, such as Tapponier & Brace (1976), observed few shear fractures in their data, and now we mention this good point in the discussion (lines 308-309).**

6. The authors attempt to explain coalescence from a linear elastic fracture mechanics perspective, with the hypothesis that fractures near each other are more likely to grow because their fracture tip stress concentrations interact. This is introduced first in section 3.4, and further expanded upon in section 4.4. This hypothesis is not well

explained or quantified: It seems to me that such an interaction depends on the length of the fractures involved (longer fractures, larger stress intensity) and on their orientation, as well as on the exact stress fields around them (e.g., mode-II fractures have reduced and increased stresses near their tip, whereas mode-I fractures do not). So I am not sure if I understand well or agree with line 208: 'The observations match the expectations of LEFM'. Secondly, contrary to this hypothesis is the statement in the introduction that LEFM cannot explain well fracture coalescence (line 30-35), so why choose this as a framework to explain the observations on coalescence?

**In the introduction, we have modified the text to specify that "such analytical formulations struggle to describe the coalescence behavior of fracture networks as they transition from distributed, disperse networks comprised of many isolated, small fractures to more localized networks comprised of well-connected, larger fractures. This transition includes a continuum of fracture development that may be divided into three endmember fracture growth modes: 1) nucleation, 2) isolated propagation and 3) coalescence." (lines 37-41).**

**Although these analytical formations struggle to describe coalescence, they can provide insights into the potential propagation of individual fractures, and how this depends on the fracture's length, orientation and stress fields (as mentioned by the reviewer). Thus, examining the extent of the agreement between these LEFM predictions (stress intensity factor) and the experimental results is useful. We agree that the stress intensity factor is controlled by the fracture length, orientation and surrounding stress field. Precisely because fracture length controls the stress intensity factor, it also controls the local stress perturbation produced by the fracture. For this reason, analytical formations suggest that fractures perturb their local stress field to a distance on the order of their length (e.g., Scholz et al., 1993). However, the nature of this perturbation can promote or hinder fracture growth depending on the loading conditions and fracture network geometry. We have now modified the text in the results, discussion and conclusion to more clearly describe this point and how our data indicate how fractures promote or hinder the growth of neighbouring fractures (lines 226-228, 241-243, 247-248, 373-375, 395-396).**

7. Section 4.2 discusses the competition between fracture nucleation and isolated propagation. This is somewhat tedious because the authors elect to use the analogy of sandstone deformation and models designed for layered sedimentary sequences for crystalline low porosity rock. I do not feel this is very informative: Triaxial deformation of granular aggregates is very different from low porosity rock, and the step from a small-sized crystalline rock sample to a sedimentary basin feels like a leap. Most discussion is summarized in the last paragraph of this section: This competition seems adequately explained by the fracture length dependent stress intensity factors, so that a few growing fractures shield shorter (nucleating) fractures. As a suggestion, the authors could analyse the lengths of the fractures in loading-parallel direction in their data to provide a somewhat more quantitative argument here.

**We agree that aspects of the micromechanisms that operate in granular aggregates differ from that of low porosity crystalline rock. We note in the text that "Granular rocks may contain mechanical heterogeneities that concentrate shear and/or tensile stresses more effectively than monzonite, which consists of an interlocking crystalline structure with relatively homogeneous mechanical properties." (lines 268-270). However, fracture development is**

similar in these rock types in that "mechanical heterogeneities control the location of fracture nucleation and the growth of preexisting fractures." (lines 258-249). Because nucleation is linked to stress concentrations, discussion of the factors that produce stress concentrations seems germane to this paper. We have rewritten parts of the discussion to emphasize the links between the different rock types (lines 275-276, 285-286, 304). We have also removed the previous Section 4.1 to improve the conciseness and focus of the discussion.

We note that in the reviewer's JGR 2020 paper (Aben et al., 2020), the authors also link experiments on low porosity crystalline rock to deformation mechanisms in porous rocks in the discussion section: "Porous rock such as sandstone is known to collapse at high hydrostatic pressure (e.g., Wong & Baud, 2012), and a similar type of local pore collapse may also occur in pulverized rock". We also follow this approach of linking mechanisms between various rock types when it is justified.

Aben, F. M., Doan, M.-L., & Mitchell, T. M. (2020). Variation of hydraulic properties due to dynamic fracture damage: Implications for fault zones. Journal of Geophysical Research: Solid Earth, 125, e2019JB018919. https://doi.org/10.1029/2019JB018919

8. I feel that Section 4.3 contains some similar problems as discussed above: The sandstone analogy is not a very helpful argument to explain competition between isolated fracture propagation and fracture coalescence in crystalline rock, and neither is fault damage zone evolution: The presented data is on an initially intact sample without a pre-existing fault zone. Further irrelevant excursions include the 316-318 on dilation in gouge materials.

In these discussion sections, we try to be careful to address the differences between the experiments analysed here and the previous work. The link between the current work and studies with sandstone and damage zones is that "observations indicate that the magnitude of confining stress influences fracture development" (line 306). Our general view is that rock deformation analyses benefit from reasonable generalization between different rock types, rather than only narrowly focusing on one rock type. Similarly, we included the description of dilation within gouge material in the section on dilatant hardening because gouge-filled fault zones are another system with dilatant hardening operates. We have rewritten portions of the discussion to improve conciseness.

9. Section 4.3 contains, in my opinion, the most interesting discussion: The influence of fluids on microfracture evolution prior to sample-sized failure. The authors first discuss the different confining pressures on all three samples as the source for the different microfracture evolution, and rule this out as a conclusion. Second, the chemical effect of water on crack propagation is discussed, followed by a discussion on the mechanical effect of a pressurized fluid. The authors conclude, rightly so, that these last two effects cannot be distinguished from each other and future research is necessary. I largely agree with the line of thought and the conclusion of the authors, but there are a few caveats and/or additional points that need to be addressed, starting at the level of the experiment: How does the sample size (4x10 mm) influence the reproducibility of the experiments, especially given the relative large (450um on average) grain size of the material (for instance, some grains seem to have dimensions of > 1mm in the 3D CT models)? Secondly, dilatancy hardening is presented as a mechanism to influence microfracture evolution, but how specifically is not clear. Dilatancy is often discussed on the scale of cm-size (and larger) shear

fractures or fault planes, where fault roughness and microfractures around the shear plane accommodate the dilation. Here, the microfracture regime does not yet have such a centralized structure, but could it be possible that larger and coalescing microfractures have a larger dilatancy rate than smaller fractures, so that the former are more affected by dilatancy? On stress corrosion: The authors could try to include a back of the envelop calculation on how long it takes for fluids to reach the crack tips during and after a deformation step – i.e., are the crack tips wetted during propagation, and has the pore fluid pressure equilibrated within the sample? I have measured hydraulic properties on these monzonites that may prove helpful to measure diffusion times. ("Variation of Hydraulic Properties Due to Dynamic Fracture Damage: Implications for Fault Zones", JGR 2020)

**Indeed, we are also curious about the reproducibility of the results, and how grain size influences potential inconsistencies. Several features are reproducible in the three experiments such as the (power law) increase of fracture volume when approaching failure. The general trends of fracture development in the present study show also that the three samples do have a similar behaviour. So, we are confident that the results are robust. We observe a variation in this behaviour for the sample that contains water (i.e., Figures 4-6). We now are careful to note that our conclusions rest on only three experiments (lines 332-335, 356-358). Experiments planned for this fall will explore a wider range of confining stresses to address this point.**

**The question of the appropriate size of the representative elementary volume (REV) in this system is critical to address but difficult to estimate. Whether or not a REV exists depends on the rheology: for elastic materials it may exist, but for softening materials it may not (*Gitman et al.*, 2007). Due to the large grain size of the monzonite relative to the core, we are close to the minimum limit of a REV in granular materials (10 grains), and far below an upper limit for stick-slip phenomena with glass beads ($10^7$) (*Evesque & Adjemian*, 2002). We now mention this good point in the text (lines 97-105).**

**Dilatancy hardening has also been observed in laboratory-sized samples, and not only systems with well-developed fault zones (e.g., *Brantut*, 2020).**

**We have now modified the text following the reviewer's good suggestion to calculate the length of time for fluid to flow across the rock core based on the porosity, and porosity-permeability calculations of Aben et al. (2020) (lines 347-356).**

**Gitman, I. M., Askes, H., & Sluys, L. J. (2007). Representative volume: existence and size determination. *Engineering fracture mechanics*, 74(16), 2518-2534.**

**Evesque, P., & Adjemian, F. (2002). Stress fluctuations and macroscopic stick-slip in granular materials. *The European Physical Journal E*, 9(3), 253-259.**

10. Maybe I have missed it, but I could not find an in-text reference to Figure 7.

    **This figure is referenced in the conclusion, and now we reference it in the discussion as well.**

11. I feel that there is some overlap in the discussion at the end of section 3.4, and the discussion on LEFM and coalescence in section 4.4. Consider cutting the part in section 3.4.

**We prefer to maintain this part of section 3.4 because the link between the analysis and the expectations of LEFM may require explanation in both places.**

12. Line 31-32: I do not think this statement is correct: LEFM is scale-independent.

    **We have rewritten this section of the introduction.**

13. Line 38: Successful in what?

    **We have rephrased the sentence for clarity (line 46).**

14. Line 40: Clarify what is meant with the mode of failure.

    **We have removed this sentence as it was extraneous.**

15. Line 60-61: Does this not depend on whether the reaction is diffusion or precipitation controlled?

    **We have rephrased the sentence accordingly (line 66-68).**

16. Line 108: The fractures have been simplified as ellipsoids, how realistic is this shape, especially for coalesced fractures?

    **For these monzonite rocks, the ellipsoidal shapes provide close approximations of the fracture geometry (i.e., Figure 1). For rocks like sandstone or marble where the grain boundaries exert a greater influence on fracture development, this approximation is further from the true shape. We have modified the text to address this good point (lines 138-140).**

17. Line 122: In the description of how fractures are tracked from one X-ray dataset to the next, would it be clearer to speak of fracture volume instead of fracture?

    **Following comment #5, we have specified that the volume (and dilation) of the fracture is critical to identifying it in the tomogram (Section 3.4). In other words, we only identify volumetric fractures, and so "fracture volume" and "fracture" are synonymous in this context.**

18. Line 126: Insert 'to' in between step and those.

    **Corrected as suggested.**

19. The figure references are somewhat chaotic: I would refrain from referring to the figures until the results section, and not refer to results figures in the introduction.

    **The end of the introduction section provides a brief summary of the analysis, including the loading conditions. And thus referencing Figures 1 and 2 here seems appropriate and helpful to the reader.**

20. Line 138, line 227: The term elastic is not correct here, because unloading at this point would not reproduce the near-horizontal stress-strain curve.

    **We agree and have removed this term.**

21. Section 3.1: How did the samples look like post-failure? Did they exhibit a single shear fracture at a 30-degree angle to the loading axis?

   **Two samples (#3 and #5) were completely crushed following macroscopic failure and so we could not recover them for further observations. For experiment #4, several scanning electron microscopy images were acquired after macroscopic failure. These images showed that one main fault, or series of connect fractures, oriented at ~30° from $\sigma_1$ formed (see Figure 4 c-d in Renard et al., 2018). We now mention this point in the text (lines 175-178).**

22. Line 147: Do I understand correctly that nucleating fractures from a previous load step are counted as propagating fractures in the next step (i.e., the nucleation counter is set to zero)?

   **Correct**.

23. Line 150: Repetition from section 2, can be removed.

   **Removed as suggested.**

24. Line 159: Was the volume of fluid expulsed from the pore pressure pumps measured? If so, do they match with the volume increase inferred from microCT?

   **We did not record the fluid expulsed from the pumps. This lack of recording was a limitation of the software that controls the HADES rig that we have now modified. For the next experiments with fluid pressure, we will record both the fluid pressure and the volume in the two pumps that independently control the pressure and flow at the inlet and outlet of the sample.**

25. Line 165: Are the exponents of the increase comparable between all three samples?

   **We have now included this result in the manuscript (lines 201-203).**

26. Line 170: develop a ! used our developed (the method was already explained in section 2).

   **We prefer to shortly describe the method here in case a reader skips the methods section.**

27. Paragraph 184-196: The technical part of how to define near and distant fractures should move to the method section. Also, how were closing and growing fractures defined?

   **Please see response to previous comment**.

28. Line 217: in during -> in.

   **Corrected as suggested.**

29. Line 322: The word 'analyses' in this context suggests some calculations/ quantification. Maybe 'Discussion'?

   **Here, we mean to refer to the analyses presented in the results (and the corresponding quantification), and not only our discussion.**

30. Figure 4: Would it be possible to indicate the four deformation stages here? Showing the fracture volume in mm3 instead of voxels would make it easier for readers to extract dilation rates. In panel (c), would it be possible to have the same scale, so that the logarithmic trends are easily comparable (also for figure 5b)?

   **Showing the 4 stages produces rather cluttered figures, so we prefer to only show the yield point on these figures. As 1 voxel is $6.5^3$ $\mu m^3$, the dilations are readily calculated using these units. We have reformatted this figure so that the plots in c) have the same scale. We also now show the fits of the exponential functions of the propagating fractures in b), from which we calculated the exponents (see comment #25).**

31. Some referencing is incomplete or skips over classic papers: line 28: The development of microfracturing with stress was already well documented by earlier studies than those cited here, especially from the 70s onward. For instance, Tapponier & Brace, 1976 have performed excellent microstructural work on this (see Paterson and Wong, section 5.7.4, for more refs).

   **We have now added additional references in the introduction (line 33-35) and throughout the text**.

32. Line 223-224: This statement is not correct: mechanical data, AEs, and microstructures show the development of microfracture networks past the yield point; Brace, Paulding, and Scholz 1966 inferred microcracking to be responsible for significant pore volume change during loading; Tapponier & Brace, 1976 and Wong 1982 show microstructures; AEs by Scholz 1968. These are examples of the older, more classic works that show this.

   **We agree that AE data and microstructural data acquired after loading support this idea**. **We have now modified the discussion by removing the section that contained this sentence. This discussion section did not significantly add to this contribution, and so we removed it for conciseness.**

33. Line 316: Martin III, 1980 ("Pore pressure stabilization of failure in westerly granite") may be more relevant here, as it shows the phenomenon in crystalline rock.

   **We have now added this very relevant reference.**

Review #2

The manuscript "The competition between fracture nucleation, propagation and coalescence in the crystalline continental crust" by Jessica A McBeck, Wenlu Zhu, and Francois Renard addresses the controls of development of fracture networks. McBeck et al. present an experimental study in which the fracture network development was assessed via microtomography during triaxial mechanical tests on two dry and one water-saturated sample. The data they acquired is remarkable and the method provides a great example of how fracture networks can be tracked during loading. The main outcome, that stress state and saturated vs dry conditions of the sample are the main controls of which (endmember) fracture network develops on the way to macroscopic failure, is not reflected in the title, introduced, clearly highlighted in the methods, represented in the results or adequately discussed. The authors need to address all of the following issues so the community can appreciate the scientific contribution.

34. The main message of the manuscript is not clear. It is not clear if they want to highlight the methodological approach or the results they obtained by applying the

method. The research question for this paper though is well hidden. The scope of the manuscript, the objectives and hypothesis are not clear.

**We wish to highlight both the methods and the results. We have significantly modified the abstract, introduction, and discussion to highlight the research questions more clearly. We also now state that the main scientific question relates to the tracking of the mode of propagation of microfractures (see answer to comment #35 below). A secondary result is the effect of pore pressure.**

35. It is not clear what the motivation for these experiments was. Data seems to be the same as in previous publications, which is fair to use as getting proposals funded and time allocated to do the experiments can be difficult, but it needs to made clear, where this data is new and where (re)used. 3. In the introduction, the overall concept of how fracture networks develop is not clearly outlined, thus that all assumptions and reasoning is vague. References are missing in many parts, which would allow substantiating some of the party awkward assumptions. The controlling variables which are used in the experiments and seem to be the main outcome are not introduced at all (effect of stress on fracturing, interstitial fluids).

   **Although the data has been described in other papers, the new contribution of this work is the method for tracking fractures such that we may classify them as nucleating, propagating or coalescing. We describe this point in the methods section 2.3 and in the introduction (lines 85-90, 144-146). We have expanded this point in the introduction section.**

   **The central characteristics of fracture network growth that we focus on in this work include the three categories of development that are described in the first sentence of the introduction, and listed in the title. There are many aspects of fracture network growth outside of the purview of this analysis that we did not describe. We have significantly modified the introduction to clarify our use of the term mode.**

   **We discuss the influence of confining stress and fluids in the discussion section in depth.**

36. The methods do not introduce the techniques applied both the mechanical loading (e.g. rate of loading) and the tomography (e.g. which voxel size), as well as how you analyse the data (e.g. volume calculation, attribution to which mode).

   **We have added these important points to the Method section 2.1. We also now describe more specifically the two other studies that described these experiments.**

37. The material used is not introduced at all. No description of the microstructure, no material properties (e.g. porosity). This makes it impossible to relate the tomography images and fracture network development to anything. The nucleation and propagation, especially at lower stress steps will be at grain boundaries and pre-existing defects and flaws.

   **We have added these important points to the Method section 2.1.**

38. The first part of the results seems to belong to the methods, yet it is not clear which point is made. The description and representation of the results are hard to follow

and do not seem to grasp/show important information. For example, you could colour the "new" fractures and the ones that coalescence in the shown steps differently.

**We assume that the reviewer is referring to section 3.1., which describes the macroscopic mechanical behaviour of the experiments. This behaviour is a result, and a method.**

**Assuming that the reviewer is referring to Figure 1, the 3 cores shown at the bottom of the figure are from the three difference experiments, and not from different steps of the same experiment. We now mention this point specifically in the caption.**

39. The structure of each section is flawed. The wording is unclear. Logical jumps make it very hard to follow the text. Especially in the introduction, the methods and results.

    **We have worked to improve wording and logic.**

40. Most parts of the discussion seem to be about something completely different than the experiment (upscaling- from 10mm to upper crustal) or research question (I am assuming crystalline rocks – yet the discussion is on sedimentary basins). The development of the fracture network is not discussed. References, if given, do not fit the topic.

    **Please see response to comments #7-8 above.**

41. The conclusion is contradicting the introduction in several aspects (e.g. LEFM) and is tedious as it simply repeats some statements made before which are not substantiated in the manuscript.

    **We have now modified the conclusion for conciseness.**

42. Detailed comments: I have commented on the manuscript in detail for the Abstract and the Introduction (see supplement .pdf). The extent of these comments highlight some of the main issues of the manuscript and are alike for the following sections. The Figures are not fitting the manuscript or provide a visualization to enhance the text, some detailed comments can be found there.

    **We have responded to all of the annotated comments in the attached document. We describe how we have modified the text in response to these comments as well.**

    Figure S1:
    - "vox" –> "voxel" - The variation in fond size and labelling position is a bit irritating. Could you work on it? - what does this # refer to? why #3, #5 and then #4. Maybe add to caption what the three panel show. - log scale hardly visible -consider using a different symbol/colour for this type to clearly distinct from the nucleation, above. - Caption: This figure does not show this. It only shows it in comparison to another figure Please name which figure this relates to. -Caption: The main trends are not indicated (in figure or text) - what are they? To make the point, you could add the trends of the 100 voxels to the figures.

    **We have reformatted this figure to improve clarity.**

    **The # refers to the experiment code number. The experiments are ordered in the figures as #3, 5, 4 because this order reflects the different loading**

**conditions. From experiment #3, 5, 4, differential stress and effective stress both increase, as stated in the caption to this figure. We are careful to include the loading conditions and fluid pressure with the # notation in all of the figures.**

**The main trends are described in the results section of the main manuscript: i.e., Figure 5, Figure 6. We now reference these figures in the caption of Figure S1.**

Figure S2: -Again, fond size and labelling position are a bit irritating. Why did you change the colour scheme? -Why is this yield (point) line in red, while in a) they are in the same colour as the other lines

**We have reformatted this figure to improve clarity. We have changed the color of the yield lines to red everywhere.**

---

## Author Comment (AC2) · 9 Sep 2020

Dear anonymous reviewer,

Thank you for these constructive reviews. We have significantly modified the abstract, introduction and discussion to more clearly specify the motivation and new contribution of this work. We respond to the comments point-by-point below. We added numbers to each point for clarity. We respond to the annotations of the manuscript in the attached document.

Thanks, Jessica McBeck

[Figure]

Review #2

34. The main message of the manuscript is not clear. It is not clear if they want to highlight the methodological approach or the results they obtained by applying the method. The research question for this paper though is well hidden. The scope of the manuscript, the objectives and hypothesis are not clear.

We wish to highlight both the methods and the results. We have significantly modified the abstract, introduction, and discussion to highlight the research questions more clearly. We also now state that the main scientific question relates to the tracking of the mode of propagation of microfractures (see answer to comment #35 below). A secondary result is the effect of pore pressure.

35. It is not clear what the motivation for these experiments was. Data seems to be the same as in previous publications, which is fair to use as getting proposals funded and time allocated to do the experiments can be difficult, but it needs to made clear, where this data is new and where (re)used. 3. In the introduction, the overall concept of how fracture networks develop is not clearly outlined, thus that all assumptions and reasoning is vague. References are missing in many parts, which would allow substantiating some of the party awkward assumptions. The controlling variables which are used in the experiments and seem to be the main outcome are not introduced at all (effect of stress on fracturing, interstitial fluids).

Although the data has been described in other papers, the new contribution of this work is the method for tracking fractures such that we may classify them as nucleating, propagating or coalescing. We describe this point in the methods section 2.3 and in the introduction (lines 85-90, 144-146). We have expanded this point in the introduction section.

The central characteristics of fracture network growth that we focus on in this work include the three categories of development that are described in the first sentence of the introduction, and listed in the title. There are many aspects of fracture network growth

outside of the purview of this analysis that we did not describe. We have significantly modified the introduction to clarify our use of the term mode.

We discuss the influence of confining stress and fluids in the discussion section in depth.

36. The methods do not introduce the techniques applied both the mechanical loading (e.g. rate of loading) and the tomography (e.g. which voxel size), as well as how you analyse the data (e.g. volume calculation, attribution to which mode).

We have added these important points to the Method section 2.1. We also now describe more specifically the two other studies that described these experiments.

37. The material used is not introduced at all. No description of the microstructure, no material properties (e.g. porosity). This makes it impossible to relate the tomography images and fracture network development to anything. The nucleation and propagation, especially at lower stress steps will be at grain boundaries and pre-existing defects and flaws.

We have added these important points to the Method section 2.1.

38. The first part of the results seems to belong to the methods, yet it is not clear which point is made. The description and representation of the results are hard to follow and do not seem to grasp/show important information. For example, you could colour the "new" fractures and the ones that coalescence in the shown steps differently.

We assume that the reviewer is referring to section 3.1., which describes the macroscopic mechanical behaviour of the experiments. This behaviour is a result, and a method.

Assuming that the reviewer is referring to Figure 1, the 3 cores shown at the bottom of the figure are from the three difference experiments, and not from different steps of the same experiment. We now mention this point specifically in the caption.

39. The structure of each section is flawed. The wording is unclear. Logical jumps make it very hard to follow the text. Especially in the introduction, the methods and results.

We have worked to improve wording and logic.

40. Most parts of the discussion seem to be about something completely different than the experiment (upscaling- from 10mm to upper crustal) or research question (I am assuming crystalline rocks – yet the discussion is on sedimentary basins). The development of the fracture network is not discussed. References, if given, do not fit the topic.

Please see response to comments #7-8 to the reviewer Dr. Aben.

41. The conclusion is contradicting the introduction in several aspects (e.g. LEFM) and is tedious as it simply repeats some statements made before which are not substantiated in the manuscript.

We have now modified the conclusion for conciseness.

42. Detailed comments: I have commented on the manuscript in detail for the Abstract and the Introduction (see supplement .pdf). The extent of these comments highlight some of the main issues of the manuscript and are alike for the following sections. The Figures are not fitting the manuscript or provide a visualization to enhance the text, some detailed comments can be found there.

We have responded to all of the annotated comments in the attached document. We describe how we have modified the text in response to these comments as well.

Figure S1: - "vox" –> "voxel" - The variation in fond size and labelling position is a bit irritating. Could you work on it? - what does this # refer to? why #3, #5 and then #4. Maybe add to caption what the three panel show. - log scale hardly visible -consider using a different symbol/colour for this type to clearly distinct from the nucleation, above. - Caption: This figure does not show this. It only shows it in comparison to another figure

Please name which figure this relates to. -Caption: The main trends are not indicated (in figure or text) - what are they? To make the point, you could add the trends of the 100 voxels to the figures.

We have reformatted this figure to improve clarity.

The # refers to the experiment code number. The experiments are ordered in the figures as #3, 5, 4 because this order reflects the different loading conditions. From experiment #3, 5, 4, differential stress and effective stress both increase, as stated in the caption to this figure. We are careful to include the loading conditions and fluid pressure with the # notation in all of the figures.

The main trends are described in the results section of the main manuscript: i.e., Figure 5, Figure 6. We now reference these figures in the caption of Figure S1.

Figure S2: -Again, fond size and labelling position are a bit irritating. Why did you change the colour scheme? -Why is this yield (point) line in red, while in a) they are in the same colour as the other lines

We have reformatted this figure to improve clarity. We have changed the color of the yield lines to red everywhere.

Please also note the supplement to this comment:
https://se.copernicus.org/preprints/se-2020-114/se-2020-114-AC2-supplement.pdf

**Supplement:**

5

**The competition between fracture nucleation, propagation and coalescence in the crystalline minimental upper crust**

Jessica A. McBeck1, Wenlu Zhu2, François Renard1,3

1Njord Centre, Department of Geosciences, University of Oslo, Norway

2Department of Geology, University of Maryland, College Park, U.S.A.

3University Grenoble Alpes, University Savoie Mont Blanc, CNRS, IRD, IFSTTAR, ISTerre, France

Correspondence to: Jessica McBeck (j.a.mcbeck@geo.uio.no)

Abs t. Different more of fracture growth produce fracture networks with distingtive geometric attributes that exert impore controls on the extern 
[revised manuscript text omitted]

---

## Referee Report (RR1)

Dear Dr. Niemeijer,

This manuscript presents an experimental analysis of microfracture development in a low porosity rock under dry and wet conditions and under confining pressure. The authors systematically presented the methodology and the experimental observations of the state-of-the-art technique. The authors attempt to simulate in-situ conditions in the upper crust. This is an important topic with significant implications to rock mechanics and natural fluids production. While the topic is important, the paper suffers from a few central weak points that need to be revised. As I worked on related topics, my revision is somewhat biased, and I apologize for the frequent self-citations.

Ze'ev Reches

Major comments:

1. First to the good parts. The experimental approach, procedures and observations are carefully described and explained. While the technique is non-trivial, the description also refers to previous publications as expected. This is the central core of the work, and should remain intact. The experimental observations provide a unique, quantitative perspective of rock dilation processes under in-situ conditions of the upper crust in terms of confining pressure and water presence. The methodology is most suitable for such important problem, and it is suggested to limit the introduction and interpretation to this topic. While this strength of the analysis is clear, the authors attempt to give the impression that the paper delivers more that it actually can. Some suggests are listed below.

2. One issue is reflected in the title that reads: "The competition between fracture nucleation, propagation and coalescence in dry and water-saturated crystalline continental upper crust." This needs to be revised including the related discussions of "competition" throughout the paper. Note these two main reasons.

    a. This study presents the "evolution" of microcracks in experiments, but it does not present a "competition" between processes. The "competition" point is also a major issue with the interpretation throughout the paper. To claim that two (or more) processes compete with each other, the authors have to quantify and compare the processes on the basis of mechanical quantities like stress, strain or energy. The paper presents the evolution with general statements with no mechanical analysis. In this respect, it is similar to Reches (1988) (citing myself, apologies) that described the "Evolution of fault patterns in clay experiments" in terms of time/deformation evolution of the faults without mechanical analysis. Mechanics is mentioned in the discussion in general terms as a potential interpretation for fault propagation. Later, Reches and Lockner (1994) presented a detailed stress analysis of microfracture evolution. In summary, the authors present well documented evolution history of nucleation, growth, dilation, and coalescence of microfractures, and they speculate about the controlling mechanisms. Competition is not analyzed.

    b. With all due respect, the analysis is limited to four samples of 0.4 cm diameter of rock with 0.045 cm mean grain size, and this is perfectly fine. However, claiming that these observations are valid for the "..crystalline continental upper crust" without an quantitative scaling attempt is not justified.

    c. For these two reasons, an appropriate title could be something like: "The evolution of nucleation, propagation and coalescence microfractures in dry and water-saturated crystalline rock"

3. The present experimental method is an excellent tool to monitor dilation by microfractures, and the authors clearly demonstrated this capability by the number of microfractures and the associated global dilation (Fig. 4-6). However, the present method is 'blind' to shear fractures unless they associated with dilation, for example, wing-cracks with dilating fractures at both ends of a shear microfracture. The evolution of shear microfractures was analyzed extensively by acoustic emission (Lockner and many others), as well as by thin-sections mapping of multiple rock deformation stages (e.g., Katz and Reches, 2004). The authors carefully, and correctly, use only the term fracture, which is commonly (not exclusively) applied to extension fractures, and the authors correctly did not refer to faults, joints or shear fractures in their experiments.

   a. This inherent limitation of only dilation detection by this experimental technology can be partly eliminated by mapping and inspection of the mapped microfractures. As the authors mentioned, it is expected that the microfracture will parallel Sig1, and indeed many fractures do. In addition, there are zones and fractures that are inclined 10-20 deg relatively to Sig1, and which can be interpreted as shear-zones or faults. For example, zones in lower-left of fig. 2a, and most fractures in fig. 3, and most fractures in stage III-IV of fig. 7. In this respect, the present observations are in very good agreement with the evolution presented in Fig. 5 of Katz and Reches (2004), and Fig. 5 in Reches (1988) (I could not resist the self-citations….).

   b. In continuation of the above, here is a suggestion that will be a significant contribution. Figs. 3 and 7 are schematic presentations of the dilated microfractures without scale and position in the sample. This presentation is fine as general display, but insufficient for evolution and certainly not for competition. It is suggested to use the detailed experimental data to prepare accurate maps (cross-sections) of the microfracture patterns. The experimental data will allow to produce maps with resolution of 10 microns that will be a new contribution of the evolution of microfracture networks, in addition to the global dilation in figs. 4-6. My bias is to use the mapping approach of my works mentioned above.

4. Discussion: Sections 4.1 and 4.2 in the discussion emphasize the inappropriate issue of 'competition' discussed above. This part should be revised to focus on the evolution of microfracture patters at the sub-millimeter scale. Section 4.3 is a highly speculative jump of many orders of magnitude to crustal scale without the required mechanical analysis. It dilutes the quality of the hard, important observations of the paper. It is suggested to delete section 4.3.

References:

Katz, O., & Reches, Z. (2004). Microfracturing, damage, and failure of brittle granites. Journal of Geophysical Research: Solid Earth, 109(B1).

Lockner, D. (1993). The role of acoustic emission in the study of rock fracture. In International Journal of Rock Mechanics and Mining Sciences & Geomechanics Abstracts (Vol. 30, No. 7, pp. 883-899).

Reches, Z. (1988). Evolution of fault patterns in clay experiments. Tectonophysics, 145(1-2), 141-156.

Reches, Z., & Lockner, D. A. (1994). Nucleation and growth of faults in brittle rocks. Journal of Geophysical Research: Solid Earth, 99, 18159-18173.

---

## Author Response (AR2)

Topical Editor Decision: Reconsider after major revisions (20 Dec 2020) by Andre R. Niemeijer

Comments to the Author:
Dear authors,

I have now received 2 reviews of your revised manuscript, one original and one new and both reviewers have similar comments about the work presented.
It is clear that the work presented is appreciated by both reviewers and warrants publication, but some revisions are needed to make the manuscript acceptable. Specifically, both reviewers question the statement of a competition of fracture nucleation, propagation and coalescence and rather view the process as an evolution. Additionally, both reviewers indicate that the extrapolation of the presented results to the crystalline upper crust, i.e. many other rock types, is not justified.
I encourage you to revise your manuscript according to the excellent comments and suggestions of the reviewers.

**Dear Editor Niemeijer, Dr. Aben and Dr. Reches,**

**Thank you for these helpful comments. We have made significant changes to the discussion sections, following your suggestions. Regarding the semantic arguments about the meaning of "competition", please see the responses to comments #8 and #17. We respond to your concerns point-by-point below in bolded font. We numbered the comments for clarity and indicate where we modified the manuscript with Word Document comments that contain the corresponding comment number (i.e., C#01 for comment #1).**

**Best,**
**Jess McBeck**

Review #1

I have read the revised version of the manuscript entitled "The competition between fracture nucleation, propagation, and coalescence in dry and water-saturated crystalline continental upper crust" by McBeck, Zhu, and Renard, and their responses to the reviewer comments. The authors have made substantial improvements based on the comments provided by reviewers and editor, which have improved the readability and clarity of the manuscript, particularly the aim and the conclusion. However, I have comments on some of the author's responses, primarily on the discussion section which remains overly cumbersome and distracting due to many tangents and unnecessarily cited literature. I have detailed my comments below.

Kind regards,

Frans Aben

1.  Comments 7 & 8, Section 4.1, and majority of section 4.2: On the excursions to literature on other lithologies and larger scale systems:

    I understand the reply from the authors to some degree, but I continue to feel that a disproportionate part of the discussion is devoted to this generalisation and is distracting to the story. I support the idea of the authors on: "Our general view is that rock deformation analyses benefit from reasonable generalization between different rock types", but the data presented in this study simply cannot help to achieve such a

reasonable generalization without doing (or finding in literature) similar types of analysis on different rock types. Without this, the discussion will remain qualitative and "hand-wavy" and does not contribute to informing the reader on the main aims and conclusions of the paper. This may result in an enumerating literature study rather than a focussed scientific manuscript. I suggest to keep these discussions short and concise rather than write lengthy paragraphs, and use analogous sparingly.

First, for section 4.1, the last paragraph should be placed after the 1st sentence of section 4.1; it adequately explains why isolated propagation is favourable to nucleation of smaller flaws. I understand from lines 258-276 that the authors attempt to discuss the initial flaw distributions in other rock types, and how that may influence the outcome measured on monzonites? This discussion does not provide an informative conclusion (line 268), and may at most provide a hypothesis for future experiments (line 275). I could not be convinced on how the discussion on the amount of stress concentrators in sandstones is germane to the main outcome of this section that isolated propagation of larger fractures is favourable to nucleation of smaller ones in crystalline rock.

The next paragraph continues the discussion with sedimentary volumes (with different lithologies, different scales, and different boundary conditions relative to the experiment) as an analogue to the experiment. I do not see clearly how this serves the main explanation of why isolated propagation is favourable to nucleation of smaller flaws; as the authors state, this is well explained by LEFM and examples from the LEFM literature may serve as better analogues/examples (e.g., Weibull theory).

**We have now shortened this section (4.1) to focus primarily on fracture development in crystalline rock. We have removed the discussion of crustal sedimentary sequences.**

**We continue to think that this manuscript benefits from describing the link between this work and the previous work on fracture development in sandstone. The link between these analyses is that both aim to understand the driving factors of fracture nucleation and propagation, and compare the dominance of these behaviors. In particular, we conclude section 4.1. with the concrete statement "in a given sandstone volume there will likely be a greater number of sites of significant stress concentrations than in a monzonite or granite volume, and thereby a larger number of sites suitable for fracture nucleation. Consequently, we may expect a greater dominance of nucleation in sandstone and other rocks with strong strength heterogeneity than observed in these monzonite rocks". We think it is valuable to mention how our results (the proportion of fracture propagation relative to nucleation) may differ between crystalline rock and granular rock.**

2. For section 4.2:
Line 306-316: Here, the authors describe the fairly well established evolution of macroscopic failure evolving from tensile to shear with increasing confining pressure. Why is it important to understand this, and the relative proportion of shear and tensile deformation, in light of the results that were obtained before macroscopic failure?

**We think that this topic is relevant for this study because it highlights a well-recognized link between confining stress and fracture network development. In the same way that this study finds a link between confining stress and the proportion of propagating vs. coalescing fractures, previous work has**

**observed a link between confining stress and the proportion of shear vs. tension. We have added an additional topic sentence describing this link more explicitly.**

**In addition, we think that this topic is relevant because the mode of deformation of the fracture (i.e., proportion of tension vs. shear) may help determine whether it propagates in isolation or coalesces with a neighbor. We provide further details in response to the next reviewer's comment (#3).**

3. Line 320: Why does a tensile fracture enable greater access to preexisting fractures than a shear fracture? Line 322: Why do mixed-mode fractures have a larger surface area than shear fractures? Is their roughness larger? How does that relate to the aperture?

    **We hypothesize that a tensile fracture may provide greater access to preexisting fractures because opening likely (necessarily?) increases the fracture aperture, as hinted by the reviewer. We have modified this paragraph accordingly to specify the link more clearly.**

4. Line 323: I am not convinced that fault damage zones need to be mentioned here: A fault zone is a shear fracture that may be compared to the macroscopic shear fracture developed at failure in triaxial experiments. The analysed fractures in this study are all tensile, near-zero offset microfractures, so do not directly compare with a shear fault. At failure, the macroscopic shear rupture and subsequent slip will create additional microfractures surrounding the shear fault by dynamic transient stresses and slip over a rough interface, but these microfractures damage zone (or meso-fracture damage zone in the field) have few to do with the pre-failure microfractures studied here.

    **We have removed this sentence, as suggested.**

5. Line 345: Saturated gouges: These are shear systems, opposed to mode-I opening of microfractures. Gouge-filled fault systems with dilation may be described not by fracture mechanics, but by frictional processes.

    **We have removed this point, as suggested.**

6. Comment 4: What was the axial loading rate, how was axial shortening measured? Loading rate is an important parameter, as it may influence strength and whether the system will be (partially) undrained during a load step.

    **We have now added this information.**

7. Comment 9: The comment on the representative elementary volume has been addressed by the authors, but I would recommend to remove the general remarks on the existence of an REV for softening materials and the upper limit of a REV for glass beads – both are not applicable to the rheology tested here. Without these remarks, the authors already show that they have considered this problem, and support the reproducibility by previous work.

    **Corrected as suggested.**

8. On some linguistics: The authors aim to track which mode of fracture network development is dominant as a function of axial load, presenting it as a "competition". I am not convinced this should be presented as a competition, as one mode of

fracture network development naturally leads to the next: If all fractures continue propagating in isolation, at some point the fracture population will have grown to fracture lengths where it is not possible anymore to stay within isolation, and all fractures are near to each other. This eliminates the mode of propagation-in-isolation. Similarly, when sufficient fractures have reached a substantial length, nucleation of smaller fractures becomes unfavorable as the longer fractures "shadow" them (this is all explained in the manuscript as well). Thus, in my view, rather than a competition between modes, it is an unavoidable sequence of modes as a function of load that have some intervals of differential stress in which both modes may contribute to fracture network evolution before one of the modes is eliminated by evolving geometrical properties (fracture length, fracture spacing). This sequence seems, pardon the pun, set in stone, so that the "winner" of the "competition" is known, so is it not a sequence rather than a competition, with the main aim of quantifying in terms of load the transition from one mode to the other?

**The central focus of this paper is to "investigate the relative contributions of three endmember deformation modes to fracture network development" (line 79-80). According to Oxford Languages dictionary, a competition is "the activity or condition of striving to gain or win something by defeating or establishing superiority over others". So a competition can be defined as any situation in which the expression of one behavior/characteristic limits the expression of another. Because we categorize the modes of fracture growth into three non-overlapping modes, the success of one limits the success of the others. For example, if a fracture is propagating in isolation, it necessarily is not coalescing. We agree that one fracture can transition between these modes, but it need not experience all three, as suggested by the reviewer. Our experiments show that a fracture can nucleation and then grow in isolation, but never coalesce with another fracture. We are thus interested in how fractures transition between the modes, and the evolving dominance of these modes. Because the modes are non-overlapping, at a given point in time in an experiment, we are able to quantify which fracture development mode is the most dominant, and thus is "winning" the competition. Thus, the sequence from nucleation to propagation to coalescence may also be framed as a competition with a different winner at various stages of deformation/differential stress. And this analysis further demonstrates that this winner changes due to differential stress and the inclusion/exclusion of fluids. Thus we think it is appropriate to frame this analysis, and comparison of the dominance of varying modes of fracture growth, as a competition.**

9. Line 24: shortly before failure close to the peak stress

**Corrected as suggested.**

10. Line 9: Specify what behavior. Also, state in the first sentence that the paper looks at fracture development in crystalline rock.

**This sentence states: "The continuum of behavior that emerges during fracture network development may be categorized into three endmember modes: fracture nucleation, isolated fracture propagation, and fracture coalescence.". Thus, the type of behavior is listed in detail after the colon.**

**We have now modified the sentence to specify that this work focuses on fracture development in crystalline rock.**

11. Line 37: The word "struggle" implies to a reader that LEFM tries to describe interaction between fractures, but fails at it. Since LEFM does not attempt to describe this at all, I suggest to replace it by "does not".

**Corrected as suggested.**

12. Line 38-39: The transition from dispersed to localized networks: This is not very clear, and may need some additional explanation. First, what is the driving force for evolving a network of fractures (e.g., continuous deformation, thermal cracking, etc), and which one will this study target? Second, how can a distributed disperse network become a localized network of connected larger fractures? What happens to the smaller fractures from the dispersed state that did not develop in larger fractures, are they healed or do we zoom out to the scale of larger fractures only for the localized network, ignoring the smaller fractures?

**We have modified the manuscript to describe the process of localizing fracture networks in greater detail, and what factors control this evolution (i.e., the driving force). We only briefly describe these processes in the introduction, and go into greater depth in the discussion.**

13. Line 85: Methods –> method

**Corrected as suggested.**

14. Line 193: stage VI stage IV

**Corrected as suggested.**

15. Line 231-234: This part of the data analysis should be mentioned in section 2 (method section).

**We prefer to describe the specifics of this analysis here, rather than only in the methods section as the reader could forget these specifics by the time they reach the results.**

16. Line 245: Be aware of the positive feedback through fracture length, which essentially does not allow for "far" fracture couples to exist anymore!

**We have now described this caveat explicitly, as suggested.**

Review #2

Dear Dr. Niemeijer,

This manuscript presents an experimental analysis of microfracture development in a low porosity rock under dry and wet conditions and under confining pressure. The authors systematically presented the methodology and the experimental observations of the state-of-the-art technique. The authors attempt to simulate in-situ conditions in the upper crust. This is an important topic with significant implications to rock mechanics and natural fluids production. While the topic is important, the paper suffers from a few central weak points that need to be revised. As I worked on related topics, my revision is somewhat biased, and I apologize for the frequent self-citations.

Ze'ev Reches

Major comments:
First to the good parts. The experimental approach, procedures and observations are carefully described and explained. While the technique is non-trivial, the description also refers to previous publications as expected. This is the central core of the work, and should remain intact. The experimental observations provide a unique, quantitative perspective of rock dilation processes under in-situ conditions of the upper crust in terms of confining pressure and water presence. The methodology is most suitable for such important problem, and it is suggested to limit the introduction and interpretation to this topic. While this strength of the analysis is clear, the authors attempt to give the impression that the paper delivers more that it actually can. Some suggests are listed below.

17. One issue is reflected in the title that reads: "The competition between fracture nucleation, propagation and coalescence in dry and water-saturated crystalline continental upper crust." This needs to be revised including the related discussions of "competition" throughout the paper. Note these two main reasons.

    This study presents the "evolution" of microcracks in experiments, but it does not present a "competition" between processes. The "competition" point is also a major issue with the interpretation throughout the paper. To claim that two (or more) processes compete with each other, the authors have to quantify and compare the processes on the basis of mechanical quantities like stress, strain or energy. The paper presents the evolution with general statements with no mechanical analysis. In this respect, it is similar to Reches (1988) (citing myself, apologies) that described the "Evolution of fault patterns in clay experiments" in terms of time/deformation evolution of the faults without mechanical analysis. Mechanics is mentioned in the discussion in general terms as a potential interpretation for fault propagation. Later, Reches and Lockner (1994) presented a detailed stress analysis of microfracture evolution. In summary, the authors present well documented evolution history of nucleation, growth, dilation, and coalescence of microfractures, and they speculate about the controlling mechanisms. Competition is not analyzed.

    With all due respect, the analysis is limited to four samples of 0.4 cm diameter of rock with 0.045 cm mean grain size, and this is perfectly fine. However, claiming that these observations are valid for the "..crystalline continental upper crust" without an quantitative scaling attempt is not justified.

    For these two reasons, an appropriate title could be something like: "The evolution of nucleation, propagation and coalescence microfractures in dry and water-saturated crystalline rock"

    **The use of the term competition is justified in this work because we categorize fracture network development into non-overlapping modes. Describing the varying dominance of these modes as a competition does not require that we compare them in terms of stress, strain, or energy. Instead, we compare them in terms of fracture volume and number, and thus use a well-constrained quantity, rather than the influence of these fractures on the internal stress, strain or energy field, which necessitate far wider error bars than our calculations of fracture volume and number.**

    **In addition, please see the response to comment #8 of the first reviewer above.**

    **We have modified the title accordingly by removing "continental upper crust" so that we do not imply that we have considered the influence of scaling.**

18. The present experimental method is an excellent tool to monitor dilation by microfractures, and the authors clearly demonstrated this capability by the number of microfractures and the associated global dilation (Fig. 4-6). However, the present method is 'blind' to shear fractures unless they associated with dilation, for example, wing-cracks with dilating fractures at both ends of a shear microfracture. The evolution of shear microfractures was analyzed extensively by acoustic emission (Lockner and many others), as well as by thin-sections mapping of multiple rock deformation stages (e.g., Katz and Reches, 2004). The authors carefully, and correctly, use only the term fracture, which is commonly (not exclusively) applied to extension fractures, and the authors correctly did not refer to faults, joints or shear fractures in their experiments.

This inherent limitation of only dilation detection by this experimental technology can be partly eliminated by mapping and inspection of the mapped microfractures. As the authors mentioned, it is expected that the microfracture will parallel Sig1, and indeed many fractures do. In addition, there are zones and fractures that are inclined 10-20 deg relatively to Sig1, and which can be interpreted as shear-zones or faults. For example, zones in lower-left of fig. 2a, and most fractures in fig. 3, and most fractures in stage IIIIV of fig. 7. In this respect, the present observations are in very good agreement with the evolution presented in Fig. 5 of Katz and Reches (2004), and Fig. 5 in Reches (1988) (I could not resist the self-citations….).

In continuation of the above, here is a suggestion that will be a significant contribution. Figs. 3 and 7 are schematic presentations of the dilated microfractures without scale and position in the sample. This presentation is fine as general display, but insufficient for evolution and certainly not for competition. It is suggested to use the detailed experimental data to prepare accurate maps (cross-sections) of the microfracture patterns. The experimental data will allow to produce maps with resolution of 10 microns that will be a new contribution of the evolution of microfracture networks, in addition to the global dilation in figs. 4-6. My bias is to use the mapping approach of my works mentioned above.

**We acknowledge that this segmentation method highlights fractures with aperture above or at the spatial resolution of the tomogram, and thus may miss some fractures that primarily host shear, with sufficiently small apertures. We mention this point in Section 2.3. We are interested in the proportion of tension vs. shear and thus have performed digital volume correlation on tomograms from our experiments (e.g., McBeck et al., 2020). This method provides tighter constraints on the proportion of tensile vs. shear strain than the fracture orientation-focused analysis suggested here.**

**In other past work, we have been more specifically interested in the orientation of fractures in these experiments. Accordingly, we have described these evolutions in previous work (Renard et al., 2018). In particular, Renard et al. (2018) track the orientation of fractures and find that they evolve toward 60 degrees from the maximum compression direction (Figure 7). Renard et al. (2018) also provide detailed maps of the fracture patterns of these experiments.**

**In addition, constructing these fracture maps would not help answer the central question of this paper of how the three endmember modes of fracture growth vary in dominance throughout loading, and vary due to confining stress and the exclusion/inclusion of fluids.**

**Renard, F., Weiss, J., Mathiesen, J., Ben-Zion, Y., Kandula, N., & Cordonnier, B. (2018). Critical evolution of damage toward system-size failure in crystalline rock. Journal of Geophysical Research: Solid Earth, 123. https://doi.org/10.1002/ 2017JB014964**

**McBeck, J., Ben-Zion, Y., & Renard, F. (2020). The mixology of precursory strain partitioning approaching brittle failure in rocks. *Geophysical Journal International, 221*(3), 1856-1872.**

19. Discussion: Sections 4.1 and 4.2 in the discussion emphasize the inappropriate issue of 'competition' discussed above. This part should be revised to focus on the evolution of microfracture patters at the sub-millimeter scale.

    **We have revised these sections following this comment, and the comments of the first reviewer (#1-5).**

20. Section 4.3 is a highly speculative jump of many orders of magnitude to crustal scale without the required mechanical analysis. It dilutes the quality of the hard, important observations of the paper. It is suggested to delete section 4.3.

    **Section 4.3 includes only one sentence that describes work focused on km-scale faults. This section, and corresponding analysis, are concerned with how nearby fractures can perturb the local stress field and influence fracture growth. Thus, this one reference is applicable as it describes the influence of fault spacing on earthquake arrest. In the same way that we observe a relationship between fault spacing and fracture growth, previous work has observed a relationship between fault spacing and fracture dynamics. We now have added a sentence to more clearly specify this link.**

---

## Author Response (AR3)

**Dear Editor Niemeijer and Dr. Aben,**

**We have provided additional explanation in the text to clarify how larger apertures could provide greater access to preexisting fractures (lines 305-7). In particular, we reason that larger apertures could lead to larger fracture surface area, and thus larger available area to which other fractures can link. We agree with the point that when the load is removed, the aperture of tensile fractures could decrease as fractures close. However, as we only examine fractures under load, this point is not germane to this study. We have now worked to correct this confusion following the suggestion of the reviewer to state more clearly that we are considering the aperture, and corresponding surface area, to make this inference.**

**Thanks,**
**Jess McBeck**

Comments from Reviewer #1:

The new version of the manuscript has a greatly improved and streamlined discussion section.
There is one argument in the discussion section that I still do not fully understand, and the authors may wish to clarify this (detailed below).

Line 303: I still do not fully understand the argument made here (initial comment and reply below), which the authors may wish to clarify. Tensile fractures indeed have a larger aperture at lower confining pressure and a larger aperture compared to shear or mixed mode fractures of the same size. But how does this provide greater access to preexisting fractures? On both sides of the tensile fracture, the solid material (including the preexisting fractures) is displaced sideways (with an axial load), and material on the top and bottom are displaced vertically (the fracture 'bulges' open), but the solid mass is not removed. When the load on the tensile fracture is removed, the aperture is reduced again and the fracture walls align (in a perfect elastic situation) – this would be akin to the geometry of a shear fracture with the same amount of fracture surface area as the tensile fracture, crosscutting the same amount of preexisting fractures. I believe some of the confusion arises from how shear and tensile fracture are compared here: Is it length (either in relaxed or in stressed conditions), fracture surface area, or aperture (in loaded conditions)?
previous comment #3: Line 320: Why does a tensile fracture enable greater access to preexisting fractures than a shear fracture? Line 322: Why do mixed-mode fractures have a larger surface area than shear fractures? Is their roughness larger? How does that relate to the aperture?
previous reply: We hypothesize that a tensile fracture may provide greater access to preexisting fractures because opening likely (necessarily?) increases the fracture aperture, as hinted by the reviewer. We have modified this paragraph accordingly to specify the link more clearly.

Kind regards,
Frans Aben